# An aircraft gas chromatograph-mass spectrometer System for Organic Fast Identification Analysis (SOFIA): design, performance and a case study of Asian monsoon pollution outflow

Efstratios Bourtsoukidis, Frank Helleis, Laura Tomsche, Horst Fischer, Rolf Hofmann, Jos Lelieveld, Jonathan Williams

Max Planck Institute for Chemistry, Mainz, 55128, Germany

*Correspondence to*: Efstratios Bourtsoukidis (e.bourtsoukidis@mpic.de)

**Abstract.** Volatile organic compounds (VOC) are important for global air quality and oxidation processes in the troposphere. In addition to ground-based measurements, the chemical evolution of such species during transport can be studied by performing in-situ airborne measurements. Generally, aircraft instrumentation needs to be sensitive, robust and sample at higher frequency than ground based systems while their construction must comply with rigorous mechanical and electrical safety standards. Here, we present a new System for Organic Fast Identification Analysis (SOFIA), which is a custom built fast Gas Chromatography – Mass Spectrometry (GC-MS) system with a time resolution of 2-3 min and the ability to quantify atmospheric mixing ratios of halocarbons (e.g. chloromethanes), hydrocarbons (e.g isoprene), oxygenated VOCs (acetone, propanal) and aromatics (e.g. benzene, toluene, butanone) from sub-ppt to ppb levels. The relatively high time resolution is the result of a novel cryogenic pre-concentration unit which rapidly cools (~6 ºC/s) the sample enrichment traps to -140 ºC, and a new chromatographic oven designed for rapid cooling rates (~30 ºC/s) and subsequent thermal stabilization. SOFIA was installed in the High Altitude and Long Range Research Aircraft (HALO) for the Oxidation Mechanism Observations (OMO) campaign in August 2015, aimed at investigating the Asian monsoon pollution outflow in the tropical upper troposphere. In addition to a comprehensive instrument characterization we present an example monsoon plume crossing flight as a case study to demonstrate the instrument capability. Hydrocarbon, halocarbon and oxygenated VOC data from SOFIA are compared with mixing ratios of carbon monoxide (CO) and methane ($CH_4$), used to define the pollution plume. By using excess (ExMR) and normalized excess mixing ratios (NEMRs) the pollution could be attributed to two air masses of distinctly different origin, identified by back-trajectory analysis. This work endorses the use of SOFIA for aircraft operation and demonstrates the value of relatively high-frequency, multicomponent measurements in atmospheric chemistry research.

## 1 Introduction

Despite their generally low ambient concentrations, organic trace gases can have significant impacts on atmospheric chemistry (Williams, 2004). Halogenated organic compounds are capable of destroying both tropospheric and stratospheric ozone (Molina and Rowland, 1974; Read et al., 2008; Saiz-Lopez et al., 2012; Wang et al., 2015), and thus altering the oxidative capacity of the atmosphere (Parrella et al., 2012; Baker et al., 2016), with prominent examples being the chlorofluorocarbons (CFCs) and the hydrochlorofluorocarbons (HCFCs). Their industrial replacements, i.e. hydrofluorocarbons (HFCs), may have lower stratospheric ozone depletion potentials but they can act as potent greenhouse gases (Velders et al., 2009). In addition, oxygenated

volatile organic compounds (OVOCs) and simple hydrocarbons (HCs), whose emissions are associated with both biogenic and anthropogenic sources, play a central role in the production and destruction of key atmospheric oxidants such as the hydroxyl radical (OH) (Kley et,. al., 1997; Atkinson, 2000; Monks et. al., 2005; Lelieveld et al., 2016) and ozone (Pusede and Cohen, 2012). It is therefore essential to monitor their atmospheric abundance and understand their source-sink dynamics particularly in the chemically sensitive but poorly accessible region of the upper troposphere and lower stratosphere.

For over sixty years, organic trace gases in the atmosphere have been measured by acquiring air samples in the field, using pressurized metal/glass containers or on adsorbent filled tubes, and subsequently measuring them "off-line" in the laboratory. A commonly used analytical technique for such samples has been gas chromatography (GC), coupled to detectors such as the mass spectrometer (MS), the flame ionization detector (FID) and the electron capture detector (ECD) (e.g. Haagen-Smit et al., 1953; Colman et al., 2001; Colomb et al., 2006; Williams et al., 2007; Pollmann et al., 2008; Lerner et al., 2017). While this approach allows for straightforward sampling in remote regions, and in the upper troposphere using research aircraft or balloons, the sample frequency is limited by the number of containers available and it is prone to interference from surface related artefacts (especially for the most reactive species (Plass-Dülmer et al., 2006)). To achieve higher time resolution and hence data density in atmospheric measurements, several "on-line" instrumental techniques have been developed based on ionization mass spectrometry (e.g. Arnold and Hauck, 1985; Crutzen et. al., 2000; Sprung et al., 2001; Blake et al., 2006; Le Breton et al., 2012). These methods use positively or negatively charged species to ionize the substances of interest prior to their detection in a mass spectrometer. While these techniques enable high resolution sampling (typically 1 s – 1 min), as they do not require a pre-concentration or separation, they suffer from lower molecular specificity compared to GC-MS, as multiple species may contribute to a given mass signal. Therefore, provided it can be made to measure fast enough, gas chromatography (GC) coupled with a mass spectrometer (MS) can be an extremely powerful tool in atmospheric research.

In recent years, there has been considerable focus on optimizing GC analysis for speed (Mastovska and Lehotay, 2003). This has led to a new generation of fast GC systems that can be applied in airborne research since they combine high time resolution and mass selectivity. Apel et al. (2003) developed the first fast GC-MS system for airborne measurements of VOCs and halocarbons (Trace Organic Gas Analyzer (TOGA); see also Apel 2016, Hornbrook et al., 2011; Apel et al., 2010, 2012).  TOGA uses a custom built liquid nitrogen ($LN_2$) fueled system to cool sample enrichment traps down to -130 °C (Apel et al., 2016) enabling a sample every 2-5 min. To our knowledge, three other fast GC systems have since been designed for aircraft operation: GhostMS (Sala et al., 2014), HCG (Jäger et al. 2014) and µDirac (Gostlow et al., 2010). All three do not require the use of a liquid cryogen and have been successfully operated in aircraft, Zeppelin or balloon campaigns. The use of $LN_2$ for cryogenic trapping has both advantages and disadvantages that are discussed in section 4.

The objective of this paper is to present a comprehensive description of the new fast GC-MS instrument, called "System for Organic Fast Identification Analysis (SOFIA)", designed for airborne measurements. In section 2 a detailed description of the major components and sampling characteristics is provided along with the instrument specifications, as defined under laboratory and field conditions. In section 3 we demonstrate the capabilities of SOFIA by presenting an example from the OMO aircraft campaign, flying over the Arabian Peninsula in August 2015 to intercept pollution plumes convected to the upper atmosphere in the Indian monsoon system. We demonstrate the usefulness of fast GC-MS measurements by studying the mixing ratios of selected chloromethanes (chloromethane ($CHCl_3$), dichloromethane ($CH_2Cl_2$), chloroform ($CHCl_3$) and carbon tetrachloride ($CCl_4$)), hydrocarbons (isoprene ($C_5H_8$), benzene ($C_6H_6$) and toluene ($C_7H_8$)), OVOCs (propanal

($C_3H_6O$) and acetone ($C_3H_6O$)) and sulphur containing species (carbon disulphide ($CS_2$)) in a pollution plume, and discuss the technical and scientific implications.

## 2 Material and methods

The system described below has been designed and constructed for use on-board the German High Altitude Long Range aircraft (HALO), a Gulfstream G550 aircraft, operated by the German Aerospace Organization (DLR). Operation on-board a high flying aircraft such as HALO requires that the analytical system can perform reliably over the widely varying conditions of external temperature (40 °C to -60 °C) and pressure (1000 mb – 200 mb). In addition, the internal cabin conditions also change and all relevant parameters must be measured for subsequent performance assessments. In the case of the SOFIA system, an additional challenge arises through the use of a liquid nitrogen ($LN_2$) cryogenic system, because the $LN_2$ evaporation rate in the container is strongly pressure and temperature dependent, thus safe and accurate pressure control within the cryogenic system is essential. The following sections present a detailed description of the major components of the SOFIA GC-MS system. They comprise of a sampling system overview (section 2.1), a detailed description of the cryogenic trapping method (section 2.2) and chromatographic GC oven (section 2.3), a characterization of the detection and quantification unit using standard gas experiments (section 2.4), a weight characterization (section 2.5) and a description of the process control and software (section 2.6). Section 2.7 presents laboratory experiments while the last part (section 2.8) reports the analytical specifications during the example flight presented in the "case study" section, which was part of the first field deployment of the instrument.

### 2.1 Sampling overview

A schematic overview of the SOFIA system is presented in **Fig. 1**. Air is drawn by the inlet pump (described in section 2.1.1) through the sampling line which extends outside the aircraft fuselage. The sample pre-concentration then follows in three stages (described in section 2.2; illustrated in supplementary **Fig. S1**). First, the air is pumped through the cooled water trap where $H_2O$ is selectively removed from the airstream and then the low temperature enrichment trap where VOC molecules are retained.  During sampling, a flow controller (**Fig. 1**; SampleFC) is used to regulate the sampling flow, and the pressure change inside a sampling volume unit (**Fig 1**; Calibrated volume) is used for accurate sample volume determination. In the second stage, the enrichment trap is heated and the volatiles are transferred to the cryofocus trap which is a narrow (0.25 mm) inert chromatography column with low volume to optimise the subsequent injection. In the third stage, the cryofocus trap is heated rapidly and the sample is injected into the chromatography column housed in the oven (denoted GC in **Fig. 1**) where it is separated prior to ionization and detection of the peak sequence by the mass spectrometer (denoted MS in **Fig. 1**).

### 2.1.1 Inlet system

Air was drawn into the aircraft through a forward-facing Trace Gas Inlet (TGI; Enviscope GmbH) (**Fig 2**, Wendisch et. al., 2016). The TGI body is constructed by aluminium (WL.-Number 3.4364, T7351) and continuously heated to 40ºC. Inside the TGI, PFA tubes (1/2' for the horizontal external tube and ¼' for the perpendicular tube) are used for the air streams. The main air flow is parallel to the flight direction and the inlet pump of SOFIA draws air from the T-piece inside the TGI, perpendicular to the flight direction. In the cabin, a heated Teflon line (¼' (0.635 cm), 2 m length, 40 ºC) brings the air into the inlet valve (**Fig. 1**). The inlet system of SOFIA was designed to compress ambient air to standard pressure at a minimum flow rate of 200 sccm in

~15 km altitude. The inlet pump and flow controller pressurize the sample to enhance the throughput in the inlet tubes (200 sccm) over the actual sampling flow (40 sccm) (**Fig. S1**). Under such conditions a maximum residence time of 15 sec can be maintained at the highest flight altitudes in the flow loop from the aircraft inlet through the zero and calibration system to the sampling point. The compression ratio of single stage all-PFA Teflon diaphragm pumps was not sufficient to fulfil our requirements, and dual stage pumps would push the overall system beyond the weight and size limits. Therefore, three small identical PFA pumps (NMP 850KNDC, KNF Global strategies AG) were used on the low pressure inlet in parallel and since the maximum pressure of the single stage was not enough it was backed by a fourth pump connected to the high pressure exit upstream of the sampling system (all four pumps are illustrated as a single one in **Fig. 1**).

Since even small leaks in the low pressure part of the inlet system can cause interferences, the pumps' tightness was tested prior to integration. It was found that the pump heads, as sold, leaked severely across the edge of the diaphragm. This issue was solved by fitting watch chronometer O-rings around the diaphragm of the individual pumps. The O-rings were fitted outside the original Teflon diaphragm, sealing edge of the pump heads and therefore they are not in contact with the sample. The pressure in the sampling system was maintained by a mechanical diaphragm pressure controller downstream of the sampling point (see 2.2.3).

Three gas cylinders (2 L, Luxfer, USA) are required for normal operation of SOFIA and they are located at the bottom of the instrument's rack. Helium (6.0 Westfalen AG, Germany) is used as carrier gas and a multicomponent gas standard (Apel-Riemer Environmental, Inc., USA) is used for in situ calibrations. The third cylinder contains pressurized $N_2$ (6.0, Westfalen AG, Germany) and is used for diagnostic purposes such as inlet blanks and to prevent jetfuel vapors being sampled during taxiing. In addition, the $N_2$ gas cylinder serves as zero air in regions that ambient sampling is not permitted. Blanks that have been obtained in such manner contain no peaks unless a leak or contamination of the inlet is present. Therefore the $N_2$ cylinder has proven to be a reliable addition for both inlet testing and standby operation.

### 2.1.2 Zero / Calibration unit

Directly downstream of the inlet system, ambient air was either sent directly to the sampling loop or through an oxidizing catalyst tube, depending on the 3-way zero valve (**Fig. 1**; ZV) position. The platinum bead packed catalyzer (PN 206016, Sigma Aldrich, USA) was maintained at 350 °C and the outflow used as VOC free air, since under these conditions all VOC measured were converted entirely to $CO_2$. The catalyzer does not significantly affect ambient water concentrations and so keeps sample humidity in calibration and zero modes the same as ambient conditions. In calibration mode, the calibration valve (**Fig. 1**; CV) was activated to allow a multicomponent calibration gas mixture (about 50 ppb of 79 compounds, Apel-Riemer Environmental, Inc.,USA) to enter via a 5 sccm flow controller (Bronkhorst, Germany) into the line exiting the unit. By modifying the inlet and the calibration gas mixture flows, a wide range of mixing ratios is achieved and used for the calibration curves.

### 2.1.3 Multi-position valves

During operation, two multiple position valves (**Fig 1**; Source valve, Traps valve; VICI, Germany) were operated concurrently. The Source valve is a 4-port valve that switches gaseous input either to a zero/calibration unit (including ambient air), a helium line or to a plug. The plugged position is used for diagnostic purposes, in particular for leak tests during flight preparation. When running continuously, the Source valve was set to helium source except during sampling mode. The Traps valve is a 6-port valve that connects the water, enrichment, and focus traps. The Traps valve valve switches between SAMPLE and INJECT modes. In the SAMPLE position,

preparation of the next sampling cycle and sampling is done in parallel, with the last sample driven through the isolated cryofocus trap to the GC column and the detector. In INJECT position, the enrichment part of the sampling loop is connected to the injection loop to transfer the sample to the cryofocus trap, while the water trap is purged in parallel to vent. An extended illustration of the flow paths can be found in the supplement (**Fig. S1**).

### 2.1.4 The sampling volume unit

The sample volume measurement unit is connected to the Traps valve downstream of the traps. It consists of a 100 ml flow controller, a calibration volume tank (432 ml), 2 solenoid valves (ET-2, 24 V, Clippard) to direct the sample flow into the sampling volume or bypass it, a sample diaphragm pump (vacuum pump; Pfeiffer MVP 006-4) and a NTC temperature sensor (Y3k-type thermocouple). The process controller isolates the pump from the sampling volume during sampling, monitors the pressure difference accumulated and calculates the air volume sampled, additionally taking the temperature reading into account. When not in sampling mode, the sample volume is connected to the sample pump to prepare for the next sampling cycle.

### 2.2 Cryogenic trapping

### 2.2.1 Design and implementation

The main objective of the cryogenic concentration systems is to achieve the minimum cycle time between the required temperature set-points (i.e. -160 ºC to 120 ºC) with minimum $LN_2$ consumption. The total operation time between $LN_2$ refilling is 17 h and covers the maximum flight duration (10 hours) and associated pre-flight ground tests (2 hours). The cryogenic concentration system (**Fig. 3**) is positioned on top of the $LN_2$ container. It consists of the pressure tight container-top plate with three cooler assemblies mounted on top of each other. In accordance with the target temperatures of each trap, the cryofocus trap is housed at the bottom, closest to the $LN_2$, the middle one is used for the enrichment trap and the upper one supports the water trap. Good thermal insulation of the whole arrangement is achieved by thin walled, stainless steel tubing and housings surrounded by an Aerogel powder (Silica granules, InnoDämm, Germany) filling up a powder-tight, 3D-printed elastomer enclosure.

Enrichment and cryofocus trap housings have a tube reaching down into a $LN_2$ container directly below the top plate of the container, from where $LN_2$ is drawn if the outlet pressure of the housing is ~50 mbar lower than the pressure in the container. The water trap housing takes cold $N_2$ gas off of the headspace of the container, because its cooling power demands are much smaller. The cooling tubes immersed in the $LN_2$, trap operations do not interfere with each other, because the change of liquid level is minimal on changing individual cooler power. The system can therefore transport $LN_2$ to rapidly cool down the enrichment and cryofocus traps (**Fig. 3**). This approach makes use of the latent heat energy (as opposed to cooled gas solutions), optimizing the cycle time.

The three traps are made of straight thin walled and uncoated stainless steel tubing (type 1.4301, Günther Lämmermeir OHG, Germany). The inner diameter of the water trap is 1/8''(0.318cm) and the enrichment trap 1/16''(0.159 cm). The water and enrichment traps are in contact with the sample. The cryofocus trap (1/16'') acts as the housing of a 0.25 mm inert chromatographic column (PN 160-255-10, Agilent Technologies, USA). Trap temperature measurement is achieved by use of a 100 μm (Omega Engineering, Germany) wire thermocouple attached via a thin walled PFA tube.

### 2.2.2 Liquid nitrogen container

The $LN_2$ container ($LN_2$-Badkryostat, CryoVac, Germany) has a usable volume of 10.5 L which is sufficient for ~17 h of continuous operation at a cycle time of 3.1 min. It is a vacuum isolated stainless steel device equipped with a mechanical overpressure valve and a rupture diaphragm (required for aircraft certification). Additionally, there is a pressure-tight cylindrical double-wall tube insert built into the container, which widens into a cone at the top, establishing the $LN_2$ reservoir from which the trap coolers are supplied (see **Fig. 3**).

In continuous operation, the inside pressure is maintained at ~50 mbar higher than the sampling system exhaust pressure (~1000 mbar) by the pressure regulating system. The small overpressure drives the $LN_2$ column up into the cone, where it evaporates at a rate proportional to the cone area immersed. This way, the $LN_2$ is evaporated directly below the traps, taking up the latent heat where it is actually needed, improving the efficiency of the system. With the filling valve open, the $LN_2$ column remains at the $LN_2$ level, which determines the idle $LN_2$ consumption. By closing the filling valve, the mechanical pressure regulating system stabilizes the system into a standby state. The level of $LN_2$ in the container is determined with capacitive $LN_2$ level probe, which is connected through a custom built voltage-frequency converter to the process controller.

### 2.2.3 Pressure regulation

The main design objectives of the pressure regulating system were a) the establishment of constant sampling pressure independent of sample air flow or outside pressure (CPOR in **Fig. 1**) and b) simple and robust control of the pressure inside the $LN_2$ system even without electrical power supplied to the instrument (Press. Reg. in **Fig. 1**). Therefore, the system was built based on pneumatic components. Since no commercial components of reasonable weight and size were available, the diaphragm pressure controllers were custom built. The diaphragms were taken from 50 mbar commercial camping gas equipment and the housing of the pressure controller had to be rebuilt. By adapting the springs, the set points and dynamic ranges could easily be adapted to our needs.

The so-called constant pressure output regulator (CPOR) is a custom made part that has a 100 cm³ reference volume attached, which automatically resets to mean ambient pressure with a time constant of 1-2 days. It is used to keep the sampling system base pressure (which is also the $LN_2$ container control reference pressure) approximately constant at a slightly higher level than ambient (~50 mbar). This pressure is monitored throughout continuous measurements to correct for possible systematic effects on overall system performance. The main uncertainty here is pressure variation caused by temperature changes, which could lead to reference pressure changes in the field on the order of 10 %. For use in SOFIA on board a research aircraft, this volume had to be isolated from changes in aircraft internal pressure (ca. 120 mbar), whereas on ground or on ships it could be opened to remove drifts caused by temperature fluctuations.

Upstream of the CPOR a differential pressure regulator (indicated as Press. Reg. in **Fig. 1; Fig 4**) of the same type was used to control both the $LN_2$ level and the standby flow with the filling valves of the $LN_2$ container closed. The filling level self-regulation operated analogously to the trap coolers. If the level in the $LN_2$ container rises, the warmer part of the wall leads to enhanced evaporation which immediately increases the pressure across the exhaust capillary, depressing the level again. The regulation of the $LN_2$ level was very important in removing the feedbacks of the regulator action on the rest of the system.

### 2.2.4 Trap temperature control

Heating of the traps is achieved via direct current from a custom built 4 V/30 A, 0-100 % DC/DC converter. The traps were electrically isolated with stainless steel fittings and Teflon ferrules. 1/16" tubing heating rates are ~100 °C/s and hard to determine accurately because of the thermocouple reading delay. 1/8" (water trap) heating rates are ~30 °C/s.

Cooling of the traps is accomplished by controlling the flow rate of $N_2$ downstream of the cooler housings by proportional valves in series with 1/16" capillaries of appropriate length. The individual trap systems are self-stabilizing, because higher $LN_2$ levels in each branch immediately lead to higher evaporation rates, thus higher pressures forcing the $LN_2$ column down again. With the current setup, cooling from +100 to a stable -180 °C is typically possible within ~50 s.

Depending on sample state, the process controller commands either the cooler valves or the trap tubes DC heater power to activate in order to maintain pre-set high or low temperatures. Typical temperature set points for water trap operation were between -40 °C and +120 °C, the enrichment trap was run between -140 °C and +120 °C and the cryofocus trap between -160 °C and +120 °C.

### 2.3 Chromatographic oven

The key factors in achieving the shortest possible cycle time of a fast GC system are a) the reproducible heating and b) extremely fast cooling to maximize the chromatogram run time while minimizing cycle time. To achieve these goals, the thermal masses of the oven, the column mandrel, the fan and the heater (**Fig. 5**) have to be minimized or eliminated. Furthermore, the surface to volume ratio of the mandrel has to be maximized and the insulation surface, mass and performance must be optimized. With the limited cooling power available for the GC oven, and to avoid condensation issues, the chromatography was run above ambient temperatures as the fan cannot create sub-ambient temperatures.

The oven was designed as a horizontally mounted toroidal structure (**Fig. 5**) to minimize internal surface area and thermal convection influence. The oven insulation is made of 3 double-walled hollow rings with wall thickness ~20 µm, making up the inner, outer and top insulation. They were manufactured by galvanizing nickel onto a 3D printed plastic core, which was dissolved afterwards by a solvent. Under this configuration the mass, heat capacity and heat conductivity were insignificant compared to convection and heat distribution.

The column mandrel (130 mm OD) consists of thin wires holding two concentric sheath metal heaters (0.1 mm * 12 mm cross section) enveloping the column in shape. The column (DB-624, 10 m, 0.25 mm, 1.4 µm; Agilent Technologies) was nested between the heaters. A thermocouple was placed on the column to get a fast response measurement of the column temperature. The sheath heaters are directly heated with a 10 V/10 A max, 0-100 % controllable custom built DC-DC converter.

The fan (EBM Papst, 56 Watts, ~ 200 mm OD) was taken out of the heated zone of the oven to remove its thermal mass. To sustain stirring of the air inside the oven and to minimize possible temperature gradients by strong thermal convection, the fan was mounted below the column, with ~ 40 % of the oven bottom area being open facing the fan blades.

For rapid cooling of the column, the oven top cover has to be opened in order to enable a high air flow rate through the ring gap around both sides of the mandrel (see **Fig. 5b**). During test experiments, it was found that the ~50 g top cover could be lifted by the airstream of the fan, resettling down to the oven side wall rings when the fan was turned down. Since it is inherently difficult to control low heat capacity / high power systems, the temperature regulator electronic parameters should be different at various temperatures to guarantee stability and

minimize stabilization time. Our custom built system is capable of controlling the system in real time. Low temperature settling time and stability were much improved by running the fan at low power along with the heater. Unfortunately, the fans implemented cannot run down to very low revolution rates to make a smooth fadeout of the fan possible, eliminating the minor disturbances on the temperature ramp (see **Fig. 5c**).

**2.4 Detector**

The GC column is connected to a quadrupole mass spectrometer MS (Agilent Technologies 5973) via a heated transfer line (143 mm deactivated column of 0.25 mm; PN 160-2255-10, Agilent Technologies). The pump of the MS was replaced by a high-power turbomolecular pump (EXT7DX, Edwards Vacuum) that can operate under higher gravitational forces to avoid problems during turbulence and on landing. The pre-pump was replaced with an oil free membrane pump (MVP006-4, Pfeifer), to avoid potential contamination. In order to achieve higher peak resolution and therefore increase precision, the electronics board of the MS was exchanged with a fast, commercial version (Sideboard PCA G3169-65015, Agilent Technologies) and the MS is operated in Selected Ion Mode (SIM) that substantially improves the detection limits while clearly separating the eluting peaks (**Table 1**). The dwell time for the individual ions selected was 10ms and the chromatographic runtime was 2.4 min (**Fig. 6**). For the laboratory characterization the temperature of the oven was programmed to start at 60 ºC, hold for 20 sec, ramp up to 70 ºC with a rate of 1.5 C/min and then ramp to 140 ºC. A dwell time of 10 ms was selected as the minimum dwell time that the chromatographic peaks were well shaped. Tests have been conducted comparing 25 ms, 10 ms and 5 ms. At 25 ms, the peaks are not clearly shaped and hence larger uncertainties are induced during peak integration. Since with 5 ms dwell time the sensitivity of the detector's response was only slightly reduced ($\approx$ 5 %), we recommend the use of 5 ms for future applications.

The linearity of the 5973MSD signal has been shown in multiple studies during the past decade (e.g. Yassaa et al., 2012). We observed a linear relationship for all species investigated ($R^2$>0.9) for mixing ratios ranging from few ppt up to 3 ppb (see supplementary **Fig. S2**). Average precision of the measurements ranges between 3.9 and 12.4 % with the respective total uncertainties is between 6.6 and 13.5 % under laboratory conditions using a multicomponent (79 species) calibration standard with mixing ratios of about 50 ppb (Apel-Riemer Environmental Inc.).The calibration standards used were within the manufacturers guaranteed accuracy period of two years. The detection limits were determined as three times the standard deviation of the signal produced by 10 zero air samples in three different concentration levels (i.e. $\approx$125 ppt, $\approx$250 ppt, $\approx$500 ppt).

**2.5 Weight**

Any instrument that is mounted on a research aircraft is subject to weight limitations. Our system is comparatively compact and light (120 kg), despite the fact that is equipped with a liquid nitrogen container. The different cryogenic methods and instrument configurations chosen for each fast GC (**Table 2**) are tailored to the specific research objectives of each instrument and come with both advantages and disadvantages concerning their application for measurements in the upper troposphere. The SOFIA instrument has been compactly built to meet the following specifications: fit one aircraft rack (65 x 55 x 163 cm) at maximum 120 kg, and operate for 17 hours on liquid nitrogen, which enables even species such as methyl chloride to be effectively trapped in-situ at an altitude range of 0-15 km.

**2.6 Process controller hardware and software**

Due to limitations in the number of instrument operators on board a research aircraft, the system was designed to be fully automated. All electronic units and sequential sampling processes are controlled with electronics

software that has been developed in-house (V25, MPIC). The V25 was additionally coupled with an external computer to trigger the MS data acquisition software Chemstation™.

Faster GC-MS cycling and therefore a higher number of chromatograms acquired creates the need for robust peak integration software. While most of the peaks could be analysed with IAU-Chrom software (Sala et al., 2014), great attention has been given to the separation of peaks that elute in very close retention times and are not clearly separated. Therefore, additional effort was put into the development of software that could clearly separate co-eluting peaks. MPIC-Chrom a new peak integration software written in IGOR, was used for the separation of acetone and propanal peaks.

## 2.7 Laboratory characterization

### 2.7.1 Water vapor

The purpose of the water trap is to retain the atmospheric water vapor that is known to induce chromatographic artefacts, degrade chromatographic separations and over time damage the column, degrade chromatographic separations and over time damage the column. It was found that large sampling flows result in poor water trapping and therefore poor reproducibility of the sampled analytes. Small flows (<40 sccm) can sufficiently remove atmospheric water vapor with up to 100 % relative humidity (RH). Supplementary figure **Fig. S2** illustrates calibration points that were obtained with both dry (0 % RH) and wet (100 % RH) air used to dilute the calibration mixture internally. To demonstrate stability of the system under changing conditions each calibration step was performed with dry and sequential wet calibrations. The detection efficiency and reproducibility of all calibration curves indicates that the water trap effectively removes water vapor during sampling.

### 2.7.2 Ozone

Problems from co-collection of ozone have been recognized previously (Goldan et al., 1995) and are documented in several studies (Bates et al., 2000; Plass-Dülmer et al., 2002; Pollmann et al., 2005; Lee et al., 2006; Apel et al., 2008; Arnts, 2008; Hellen et al., 2012). The ozone interference can be either positive via artefact formation from reaction with ozone in the sampling system (e.g. OVOCs) or negative via the loss of analytes during the enrichment stage from oxidation by ozone. The positive effect was investigated by sampling VOC-free air from an ozone generator (Thermo Environmental Inc., 49C $O_3$ generator, USA) under increasing ozone mixing ratios (0-200 ppb). Even with the highest $O_3$ mixing ratios tested, we did not observe any artefact formation for our system.

To investigate the loss of analytes from oxidation by ozone, the air produced by the ozone generator was externally mixed with the multicomponent calibration gas standard, achieving a mixing ratio of about 0.5 ppb for the investigated species. During the OMO campaign about 99 % of the ambient ozone measurements were below 100 ppb and the maximum recorded value was 114 ppb. Hence, we have conducted experiments up to 150 ppb of $O_3$. **Fig 7** illustrates the percentage difference (blue circles) from the reference value obtained at 0 ppb of $O_3$. No artefacts were observed for all species up to 75 ppb of $O_3$. However, for $O_3$ mixing ratios higher than 75 ppb, the double bonded species (i.e. isoprene, CFC-113, trichloroethylene and tetrachloroethylene) were reduced as a result of ozonolysis. The maximum loss was observed for trichloroethylene at 150 ppb $O_3$ ($\approx$20 % of the reference value). In contrast with the oxidized species, acetone and propanal were increased for $O_3$ mixing ratios higher than 75 ppb. This increase can be attributed to production by ozone reactions occurring with other

species that are present in the multicomponent gas standard, as the supply of ozone enriched zero air did not result in production of OVOCs.

To better address the observed ozone artefacts, an ozone scrubber (sodium thiosulfate ($Na_2S_2O_3$) implemented quartz filter; 47 mm, Whatman, UK) was installed the inlet. Prior to the experiments, the ozone scrubber was tested for artefacts by comparing the response of about 0.5 ppb of calibration gas mixture with and without the ozone scrubber. Since no artefacts were observed, the same experiment was performed with the ozone scrubber in line. As demonstrated in **Fig. 7** (red squares) all the investigated species displayed a similar response independent of the ozone mixing ratios. Therefore, we conclude that the use of an ozone scrubber is essential for the accurate determination of analytes under high ozone mixing ratios ($\geq$100 ppb of $O_3$).

**2.8 Specifications during OMO campaign**

The system was installed on board the HALO aircraft after its final configuration and certification. At first, a limited number of compounds (11) was monitored in order to ensure reliable quantification (**Table 1**). At the start of the OMO campaign, high sampling flows (100 sccm) resulted in inefficient water removal and hence poor and non-reproducible chromatographic peaks. The solution was to operate the system with a lower sampling flow (40sccm) and only at high altitudes where low dew point temperatures do not affect the sampling procedure. Post campaign tests (see section 2.7) showed that under such small sampling flows the water trap sufficiently retain water vapor and therefore the instrument can be operated even under ground conditions. The sampling time was 1min so a total volume of 40$\pm$6 ml was collected into the traps. During sample collection, the water trap temperature was set to -30$\pm$0.3 $^{\circ}$C and the enrichment trap to -140$\pm$4 $^{\circ}$C. During the sample transfer, the cryofocus trap was set to -160$\pm$1 $^{\circ}$C. All traps were then heated to 120 $^{\circ}$C to ensure that all volatiles were desorbed efficiently from each trap.

Prior to each flight, a pre-flight protocol was followed. The instrument was turned on (power consumption $\approx$1000 W) and all gas cylinders were opened. During the filling of $LN_2$ container ($\approx$ 10 min), the traps valve was set to INJECT position (see **Fig. S1**) and the flow controller was set to zero. This allows evacuation of any residual air in the lines and inspection of major leaks by monitoring the minimum pressure inside the MS. Subsequently, the MS heaters were turned on and the traps and GC column were heated to 100 $^{\circ}$C for about half hour, even if the MS and the heaters can reach the desired temperatures in less than 10 min. Once the aircraft was outside the hangar, the mass spectrometer was tuned and inspected for any water and air residuals in the MS. The minimum start-up time required is as low as 30 min but usually a full hour was required in order to produce reliable chromatograms, clean blanks and stable calibrations. When assured that there were no leaks in the system, a multipoint calibration was performed and the missions' sequence was set. A common strategy was to perform three calibration steps (stable mixing ratio of about 100 ppt) or three zero air measurements after 20 consecutive ambient samples. The sequence was stopped above sampling restricted areas and calibrations with dry $N_2$ compressed gas were performed. In addition, the sequence was reset in calibration mode whenever the flying altitude was changed. Upon landing and given the time restrictions that are usually present, a calibration check was performed and the MS was vented ($\approx$40 min) until the power supply from the aircraft was terminated. The GC oven temperature was programmed to start from 50 $^{\circ}$C, hold for 20 sec ramp to 80 $^{\circ}$C at 2 $^{\circ}$C/sec and then ramp to 150 $^{\circ}$C at a rate of 1 $^{\circ}$C/sec (**Fig. 5**). An ambient chromatogram with these settings is shown in supplementary figure **Fig. S3**.Under the initial configuration of the GC oven, imperfect reproducibility of the temperature ramp resulted in small retention time shifts that nonetheless remained within the selected SIM time windows. Post-campaign improvements on the oven ventilation control system and method (i.e. improved oven stability by running the fan below its commercial speed, increased initial oven temperature at 60 $^{\circ}$C) resulted in

more reproducible temperature profiles largely eliminating the retention time shifts. In the first flight campaign a sample was acquired every 3.1 minutes with an MS run of 2.4 min.. In order to assess possible pressure dependencies, calibrations were performed at each pressure level during each flight. During flight, higher uncertainty values were determined as a result of higher precision error (**Table 1**).

## 3. Case study of Asian monsoon outflow

### 3.1 Flight 20150813

The scientific aim of the OMO campaign was to investigate the oxidation processes in the convectively lifted air masses that originate in polluted areas of South Asia, and are then transported within the Indian monsoon anticyclonic flow system in the upper troposphere. The base of operation in the eastern Mediterranean was Paphos airport, in Cyprus, and flights generally headed east over the Arabian Peninsula to intercept monsoon flows heading west. Methane, measured by IR absorption spectroscopy (Schiller et al., 2008, Tadic et al., 2017) was used to identify monsoon outflow influenced air masses. A threshold methane value was derived to identify such plumes based on the average of profiles (4-10 km height; 12 flights) over Cyprus, Italy and Germany, which represented the European background without monsoon influence. The threshold was calculated as the sum of the averaged observations plus two times the standard deviation (threshold = average + 2*σ = 1879.8 ppb). Methane mixing ratios above this threshold were assumed to be influenced by the Indian summer monsoon system.

On August 13th, 2015, a flight was performed to traverse the anticyclonic system that was forecasted to extend over the eastern part of the Arabian Peninsula at high altitudes (see **Fig. 8**). The results from this flight will be used as a case study in order to demonstrate the instrument performance.

### 3.2 Pollution plume characteristics

According to the aforementioned $CH_4$ threshold, monsoon influenced air masses were mainly encountered on the eastern part of the flight track, with a few additional areas located above continental Saudi Arabia. On average, $CH_4$ increased by 66.2±23.2 ppb and CO by 29.5±12.3 ppb in the pollution plumes (see red dots in **Fig. 9a**). On the return flight, very clean air masses were encountered and these mixing ratios were considered as background (12:30-13:30 UTC). We define the excess mixing ratios (ExMR) as the difference between the observations obtained under background conditions (bg) and the respective mixing ratios measured within the pollution plume (Yokelson et al., 2013).

Increased mixing ratios of all hydro- and chlorocarbons were observed in the pollution plume except for carbon tetrachloride ($CCl_4$) which remained constant during the entire flight ($[CCl_4]_{BG}$ = 124±13 ppt, $[CCl_4]_{plume}$ = 123±14 ppt). The largest increases were observed for benzene ($ExMR_{C6H6}$ = 269 %) and acetone ($ExMR_{C3H6O}$ = 225 %) while strong increases were also observed for chloromethane ($ExMR_{CH3Cl}$ = 140 %) and chloroform ($ExMR_{CHCl3}$ = 55 %). Dichloromethane mixing ratios were higher by 18 % on average in the pollution plumes, however, this falls within the uncertainty range of the measurement on this flight. Besides the hydro- and chlorocarbons, carbon disulphide was more than double as high in the plume ($ExMR_{CS2}$ = 109 %) with a maximum measured mixing ratio of 23 ppt.

The highest correlation coefficient between the commonly used pollution marker CO and the monitored hydrocarbons was observed for benzene ($CC_{CO,C6H6}$ = 0.85) and acetone ($CC_{CO,C6H6}$ = 0.75). High correlations were also observed between benzene and chloroform ($CC_{C6H6,CHCl3}$ = 0.78) as well as benzene and carbon

disulphide ($CC_{C6H6,CS2} = 0.61$), and even higher for the air masses marked as pollution plumes ($CC_{C6H6,CHCl3} = 0.81$, $CC_{C6H6,CS2} = 0.92$).

### 3.3 Air mass separation

The highest mixing ratios of $CH_4$ and CO were observed between 09:53-10:40 UTC over Oman (**Fig. 9** and **Fig. 12**) and with relatively stable plume delineator abundance ([$CH_4$] = 1925±14 ppb ; [CO] = 109±7 ppb) which is indicative of a large scale pollution plume. Interestingly, the main species measured by SOFIA reveal markedly different mixing ratios for the first and second part of the plume. This suggests that, what appears to be one plume in the relatively unspecific marker compounds (CO and $CH_4$), has two distinctly different composition regions when volatile organic compounds are considered. To investigate the chemical differences between these air masses, the plume was subdivided into two separate plumes termed as P1 and P2.

In **Fig. 10** we illustrate the ExMRs of the main VOC species measured over P1 and P2 against the respective ExMR of $CH_4$ and CO. Benzene, acetone and chloroform increased in both plumes, moderately correlated with the respective increase in the CO mixing ratios ($R^2_{C6H6,CO} = 0.6$ , $R^2_{C3H6O,CO} = 0.61$, $R^2_{CHCl3,CO} = 0.52$).

Benzene is the most abundant aromatic hydrocarbon in the atmosphere (Martín-Reviejo and Wirtz, 2005) and its gas phase chemistry is dominated by the reaction with OH radical (Bloss et al., 2005). Differences between biomass burning and pollution outflow can be traced by the benzene/CO ratio. Scheeren et al. (2003) reported a ratio of 0.23 ppt ppb$^{-1}$ inside the monsoon outflow while Andreae and Merlet (2001) derived a ratio of 1.3 ppt ppb$^{-1}$ for biomass burning plumes. In our measurements, the benzene/CO ratio was significantly increased in P2 (Benz/CO$_{P1}$ = 0.4 ppt ppb$^{-1}$, Benz/CO$_{P2}$ = 0.7 ppt ppb$^{-1}$). In another study, Hornbrook et al. (2011) used the excess ratios (ΔBenz/ΔCO) for all biomass burning influenced air masses and reported values that range between 0.5 and 2.5 ppt ppb$^{-1}$. The respective values were increased by ~80 % in P2 (compared with P1), with an average ratio of 1.44±0.19 ppt ppb$^{-1}$ indicating the influence of biomass burning emissions in the second part of the plume.

Acetone is the most abundant OVOC in the upper troposphere with mixing ratios that can reach 2 ppb (Pöschl et al., 2001) with strong seasonal variations at the mid-latitude tropopause (Sprung and Zahn, 2010). It has both anthropogenic and biogenic sources (Jacob et al., 2002; Khan et al., 2015) but it can be also formed by the oxidation of precursor compounds such as propane (Jacob et al., 2002; Fischer et al., 2012). Biomass burning is another direct source of acetone (Holzinger et al., 2005). We have observed values up to 1.5 ppb in the Asian monsoon plume with highly elevated mixing ratios in P2 ([$C_3H_6O$]$_{P1}$ = 995±246 ppt, [$C_3H_6O$]$_{P2}$ = 1397±18 ppt), identified as biomass burning influence. At the same time, propanal was moderately increased from 19±4 ppt in P1 to 29±5 ppt in P2. In a recent study, Fischbeck et al. (2017) showed that the increase of acetone in such pollution plumes is more likely to depend on the initial mixing ratios at the source rather than on secondary production.

Chloroform has both anthropogenic and biogenic sources, while biomass burning is considered to be an important contributor (Laturnus et al., 2002). We observed an increase of chloroform in P1 ([$CHCl_3$]$_{P1}$ = 15±3 ppt) and a stronger increase in P2 ([$CHCl_3$]$_{P2}$ = 22.4±2 ppt) compared to the background measurements ([$CHCl_3$]$_{BG}$ = 10±1 ppt). In general, chloroform increased in a linear relationship with benzene.

For most species we observed enhanced mixing ratios in the second part of the plume (P2). The only compound that displayed a different tendency is chloromethane, which was higher in P1 compared to P2 ([$CH_3Cl$]$_{P1}$ = 1485±274 ppt, [$CH_3Cl$]$_{P2}$ = 988±136 ppt). While biomass burning is also a significant source of chloromethane (Rudolph et. al., 1995; Andreae and Merlet, 2001; Keppler et al., 2005; Umezawa et al., 2014), we identified stronger emissions from the non-burning monsoon outflow, which indicates that different strong sources are

present. Scheeren et al. (2003) have shown that a ratio of 10 ppt ppb$^{-1}$ is indicative of Asian pollution over the eastern Mediterranean. Hence the higher CH$_3$Cl/CO ratios observed in P1 ([CH$_3$Cl/CO]$_{P1}$ = 14.1$\pm$2.6 ppt ppb$^{-1}$) compared with P2 ([CH$_3$Cl/CO]$_{P2}$ = 8.7$\pm$1.5 ppt ppb$^{-1}$) are indicative for pollution outflow, which may have been influenced by other strong sources such as tropical vegetation (Gebhardt et. al., 2008), biofuel use (Lobert et al., 1999) or the burning of agricultural residues, waste and dung (Scheeren et al., 2002).

To further investigate the air mass characteristics, normalized excess mixing ratios (NERMs) relative to CO, were calculated and presented in **Fig. 11**. The observed NERMs between benzene and chloromethane reveal a distinct separation of the relationships between the two parts of the plume. The linear relationship for chloromethane inside P2 (R$^2$ = 0.74), in combination with the range of CH$_3$Cl/CO ratios (6.7 to 10.9 ppt ppb$^{-1}$) support the assumption that the air was influenced by biomass burning (Scheeren et al., 2003). The NERMs of chloroform and carbon disulphide increased in a similar linear manner. Especially chloroform enhancement ratios, relative to CO, were uniquely correlated with the respective enhancement of benzene (R$^2$ = 0.98; **Fig. 10**). We further examine the distinct plume characteristics by calculating 10 day back-trajectories using the FLEXible PARTicle dispersion model (FLEXPART; Stohl et al., 1998). As shown in **Fig. 12**, when HALO descended by 1 km at 10:21 UTC, a different air mass was measured. P1 originated over north India and at higher altitudes (~8km) and was transported within the prevailing anticyclonic system. In contrast, the air mass P2 emerged from lower altitudes over central India with some influence from Bangladesh, Bhutan and west Myanmar, where biomass burning (Streets et al., 2003; van der Werf et al., 2006) and fuel consumption (van der Werf et al., 2010) are more prominent.

**4. Discussion**

The need for high time resolution monitoring in the upper troposphere has given rise to a new generation of custom made, fast GC instruments that were designed to operate on board a research aircraft (**Table 2**). The most critical parameter for fast GC instrumentation on board a research aircraft is time resolution. As demonstrated in the case study section, the aircraft can rapidly cross air masses with different characteristics and high resolution monitoring is essential to interpret the underlying atmospheric phenomena.

The main constraints in achieving high time resolution sampling for GC-MS systems are the cooling and heating rates of the traps and GC oven, together with the inherent limitations of chromatographic separation time. During the first campaign (OMO), our system was operated with a time resolution of 3.1 min, while a reduction to 2 min is feasible due to the high cooling rates of both the GC oven and traps' housing. Stable and reproducible temperature are essential in to ensure reproducible retention times that ease the analysis.

Quadrupole MS has been the preferred method for quantitative detection despite the inherent restrictions on the number of species monitored due to limits in the number of measurement points needed to accurately define the fast eluting peaks and the numbers of ions that can be sequentially monitored. In the selected ion monitoring mode (SIM) several quadrupole systems have proven to be sufficiently sensitive, robust, with good reproducibility and a high degree of linearity over a wide range of mixing ratios (e.g. Apel et al., 2003). Importantly, when targeting fast measurement times the MS allows the separation of co-eluting substances that can be distinguished by their different mass and fragmentation patterns. While considerable improvements on the mass resolving power and speed could be achieved with a time-of-flight (TOF-MS) mass spectrometer that can simultaneously measure all mass-to-charge signals at high frequency, detector non-linearities of the instrument sensitivity have been observed (Hoker et al., 2015; Obersteiner et al., 2016a).

Generally for all systems, prior to detection, the substances of interest are pre-concentrated in an enrichment trap, where low temperatures are used to retain them. The range of trapping temperatures required is highly dependent on the target substances. SOFIA utilizes $LN_2$ in order to achieve large cooling capacity allowing high vapor pressure VOCs such as methyl chloride to be trapped without the need of an absorbent material that may induce artifacts and memory effects (Apel et al., 2003). The main disadvantage of using $LN_2$ as cryogen is the significant volumes required in combination with safety restrictions and supply in remote locations. The SOFIA system can be used on routine flights that do not exceed 17 h. However, multiple flights that include an extended layover or remote site landing for an overnight stay will restrict the operation of SOFIA to the outward flight, unless a re-supply of $LN_2$ is available in the host airport.

Alternative cryogen free approaches have also been realized. Compression coolers containing refrigerant (e.g. 1,1,1,2-tetrafluorethane) in combination with a tube evaporator that cools a liquid such as 50 % ethanol have been applied (Jäger, 2015). The disadvantage of this method is the reduced cooling capacity with associated limitations in retaining high vapor pressure substances. A further solution has been the use of stirling coolers as demonstrated by Obersteiner et al. (2016b) and Lerner et al. (2017). These only require electrical power and hence are eminently suitable for long term use at remote locations. The cooling rates and minimum attainable temperatures are comparable (albeit slightly higher) than with the $LN_2$ systems, even if they are not as powerful. In any case, low adsorption temperatures will result in trapping $CO_2$ which can induce chromatographic and detection problems depending on the selected ions monitored. In our system the most abundant atmospheric gases (nitrogen, oxygen, argon) will not be concentrated in the sample, but the less volatile gases such as $CO_2$ are trapped. The elution of $CO_2$ restricts the range of the analytes that can be monitored (e.g. acetaldehyde). Nonetheless, the selected species that are implemented in our method do not have interferences with the $CO_2$ ions and hence our measurements were not influenced by ambient $CO_2$.

## 5. Conclusions

We have developed a new fast GC-MS instrument for airborne measurements of volatile organic compounds including hydro- and halocarbons, OVOC and sulphur species. The system incorporates a novel cryogen-conservative VOC enrichment system that is based on the differential pressure between a $LN_2$ dewar and the trap housing to transport the cryogen as a liquid, and rapidly cool the traps to the desired temperatures. In addition, we have developed a new chromatographic oven with exceptionally high cooling rates (30 ℃/min) and rapid stabilization which helps achieve relatively high measurement time resolution. SOFIA was operated on-board the HALO research aircraft during the OMO campaign to study the convectively transported pollutants within the Indian monsoon anticyclone system. The high time resolution allowed the investigation of a seemingly uniform pollution plume and distinguish two notably different air masses. We have confirmed the distinct origins of these air masses with the use of a back-trajectory transport model, and conclude that the use of on-line, high resolution monitoring is essential for the adequate characterization of air masses and atmospheric processes that take place in the upper troposphere.

**Acknowledgments**. We would like to thank all participants of the OMO campaign, the German Aerospace center (DLR), Enviscope GmbH and EDT Offshore Ltd for the excellent cooperation during the field campaign. In particular, we thank Rolf Maser, Marcel Dorf and Kyriakos Michael for their generous support at Paphos airport in Cyprus. We thank Eric Apel and Aaron Johnson for the helpful discussions during the preliminary design phase.

**7. Tables and figures**

| Compound | Formula | SIM ion | Detection limit (ppt) | Average precision of measurements | Total uncertainty (%) | RT | RTstd |
|---|---|---|---|---|---|---|---|
| Methyl chloride | $CH_3Cl$ | 50,52 | 3 (6) | 5,2 (3,7) | 7,5(6,5) | 0,43 | 0,005 |
| Methyl bromide | $CH_3Br$ | 94 | 1 | 5,2 | 7,5 | 0,52 | 0,008 |
| Trichlorofluoromethane | $CCl_3F$ | 101 | 1 | 5,4 | 7,6 | 0,61 | 0,02 |
| Isoprene | $C_5H_8$ | 67 | 3 (6) | 6,6 (10,5) | 8,5 (11,8) | 0,68 | 0,02 |
| Propanal | $C_3H_6O$ | 58 | 11 (19) | 8,3 (5,1) | 9,9 (7,4) | 0,72 | 0,02 |
| Ethane,1,1,2-trichloro-1,2,2-trifl | $C_2Cl_3F_3$ | 101 | 1 | 3,9 | 6,6 | 0,73 | 0,02 |
| Acetone | $C_3H_6O$ | 58 | 12 (21) | 7,5 (6,1) | 9,2 (8,1) | 0,74 | 0,02 |
| Methyl iodine | $CH_3I$ | 142 | 1 | 5,1 | 7,4 | 0,76 | 0,02 |
| Carbon disulfide | $CS_2$ | 76 | 1 (4) | 5,3 (11,6) | 7,6 (12,8) | 0,78 | 0,02 |
| Dichloromethane | $CH_2Cl_2$ | 49 | 3 (9) | 5,4 (11,3) | 7,6 (12,5) | 0,84 | 0,02 |
| Butanone | $C_4H_8O$ | 43 | 10 | 7,4 | 9,2 | 1,02 | 0,02 |
| Chloroform | $CHCl_3$ | 83 | 1 (4) | 6,1 (10,4) | 8,1 (11,7) | 1,06 | 0,03 |
| 1,1,1-Trichloroethane | $C_2H_3Cl_3$ | 97 | 1 | 6,5 | 8,4 | 1,46 | 0,05 |
| Cyclohexane | $C_6H_{12}$ | 56 | 4 | 5,2 | 7,5 | 1,48 | 0,05 |
| Carbon tetrachloride | $CCl_4$ | 117 | 1 (5) | 5,9 (6,5) | 8 (8,5) | 1,52 | 0,05 |
| Benzene | $C_6H_6$ | 78 | 1 (4) | 7,7 (6,4) | 9,4 (8,4) | 1,60 | 0,03 |
| 1,2-Dichloroethane | $C_2H_4Cl_2$ | 62 | 5 | 10,6 | 11,9 | 1,62 | 0,02 |
| Trichloroethylene | $C_2HCl_3$ | 130 | 2 | 9,6 | 11,0 | 1,82 | 0,03 |
| Toluene | $C_7H_8$ | 91 | 3 (8) | 6,9 (5,2) | 8,8 (7,5) | 2,20 | 0,03 |
| Tetrachloroethylene | $C_2Cl_4$ | 166 | 3 (7) | 12,4 (10,2) | 13,5 (11,5) | 2,35 | 0,03 |

**Table 1.** Selective Ion Mode (SIM) was used to measure the listed compounds. The total uncertainty takes into account the measurement precision (derived as average of the 3 different concentration levels), the 5 % uncertainty of the standard gas and the 2 % uncertainty of the sampling volume. Retention times (RT) and their standard deviations from 2.4 min chromatograms are given in the last two columns. The values in brackets indicate the performance during flight 20150813.

| Instrument | Detector | Cryogenic method | Time resolution (min) | Adsorption temp. (°C) | Lowest DL (ppt) | Highest precision (%) | Weight (kg) | Reference |
|---|---|---|---|---|---|---|---|---|
| SOFIA | MSD 5973 | LN$_2$ | 3 (2) | -140 | 1 | 4 | 120 | This study |
| TOGA | MSD 5973 | LN$_2$ | 2 | -130 | 1 | 3 | <200 | Apel (2016) |
| GhostMS | MSD 5975 | stirling cooler | 4,3 | -100 | 0,001 | 2 | n.r. | Sala et al. (2014) |
| HGC | MSD 5975 | compressed refrigerant | 9 | 30 | 1 | <1 | 128 | Jäger (2015) |
| µDirac | ECD | n.r. | 8-15 | 15-25 | 0,5 | 1 | 11 | Gostlow et al. (2010) |

**Table 2.** Custom built fast GC instruments for on-line monitoring of organic trace gases. Non reported information is denoted with n.r.

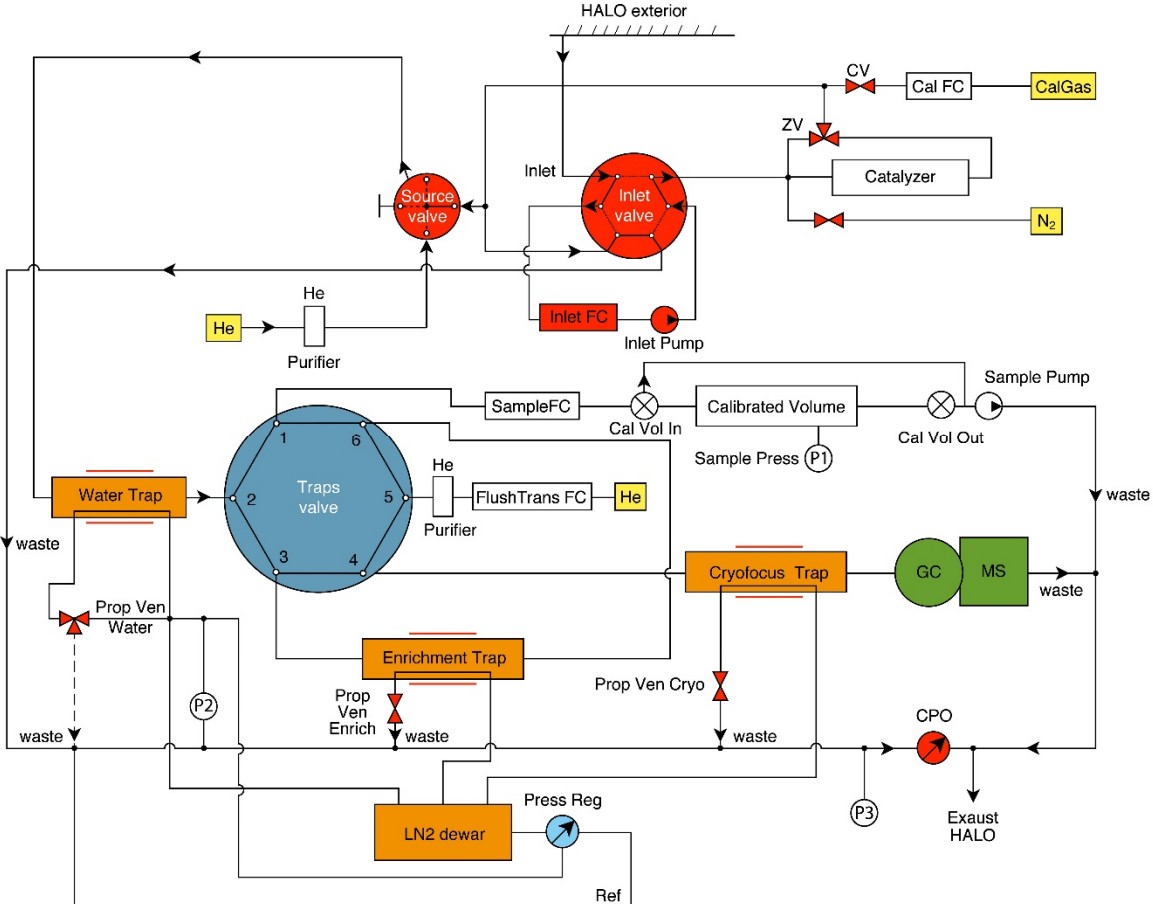

**Figure 1:** Schematic overview of SOFIA.

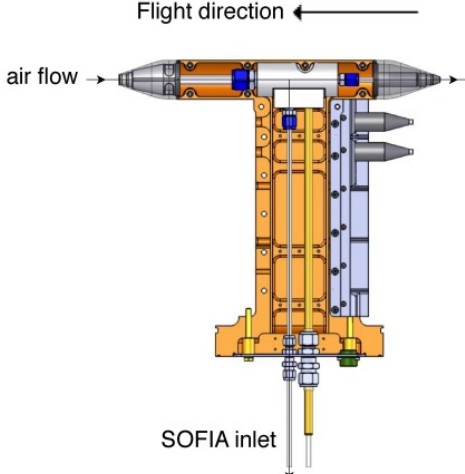

**Figure 2**. Trace Gas Inlet (TGI) installed on-board HALO.

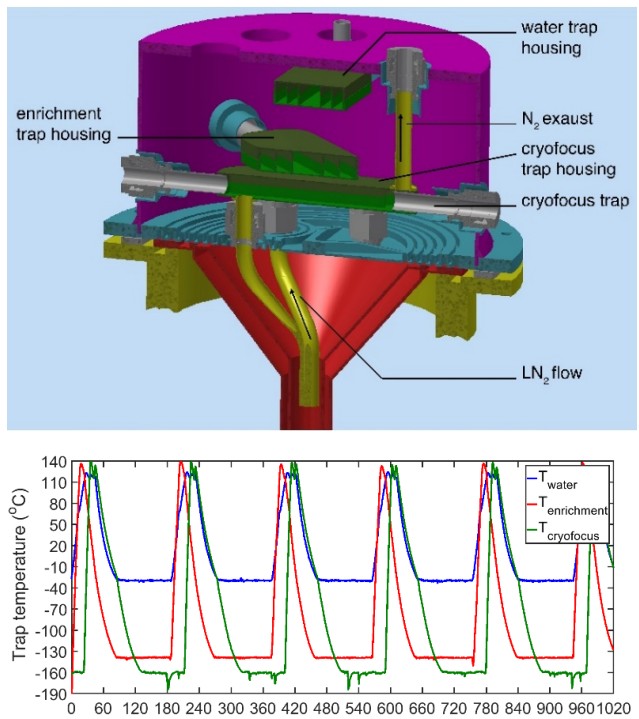

**Figure 3. Cryocooler and trap temperatures.** Schematic of the trap housing (up) and illustration of temperatures over six cycles (down). The water trap ($T_{water}$) is shown in blue, the enrichment trap ($T_{enrichment}$) in red and the cryofocus trap ($T_{cryo}$) in green.

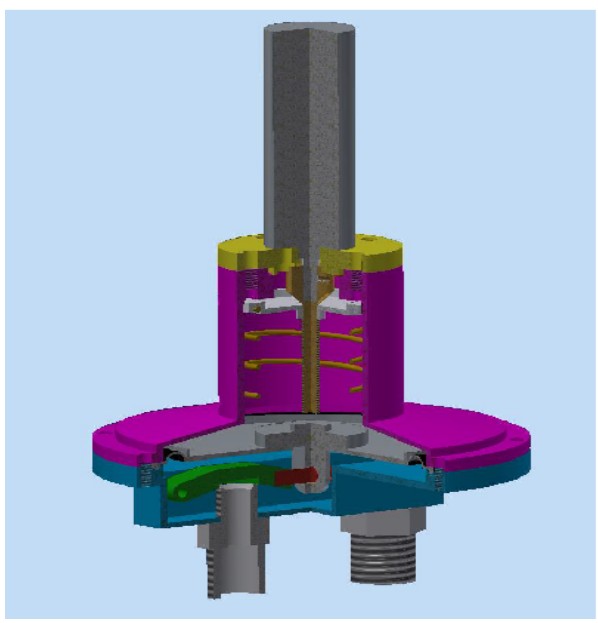

**Figure 4.** Pressure regulator of liquid nitrogen. The reference pressure volume is isolated by the purple aluminium cover. Gaseous N$_2$ evaporates from LN2 dewar and enters the bottom right inlet. Once the pressure is higher than the reference, the membrane (placed between the blue and purple part) is regulated by the green spring and the nitrogen stream is directed to the exaust (bottom left tube) releasing the pressure inside the LN2 container.

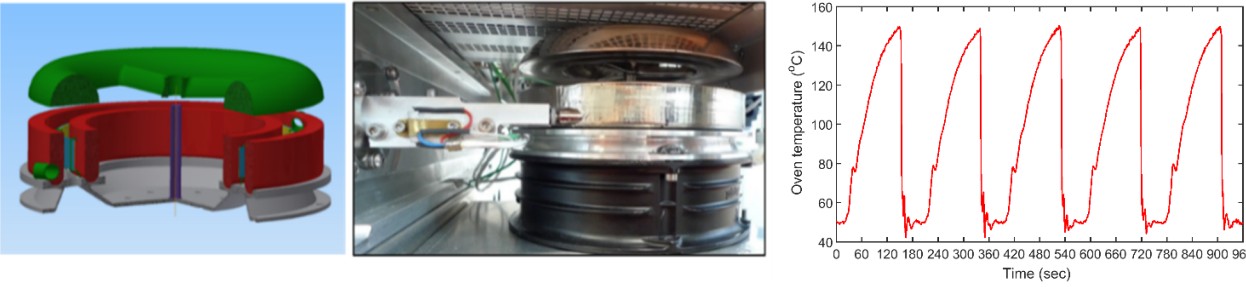

**Figure 5. The chromatographic oven.** Schematic design of the oven (left), photo of the oven during cooling (middle) and illustration of the oven temperature from six consecutive cycles during the case study flight (right).

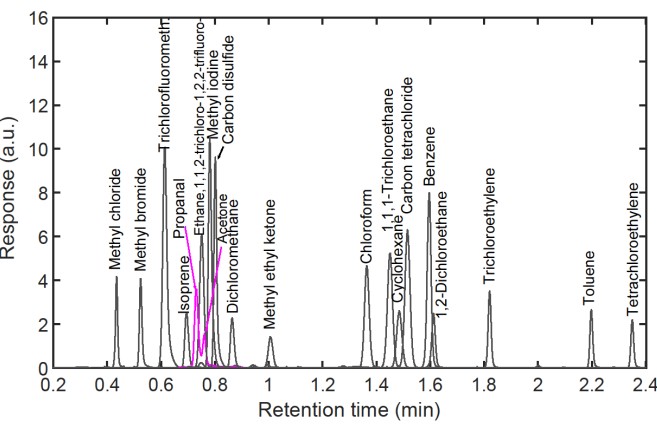

**Figure 6. Chromatogram using standard gas.** Elution times during a 3 min sample. The signals of propanal and acetone (purple) are illustrated as 10x the raw signals. The monitored compounds are listed in Table 1.

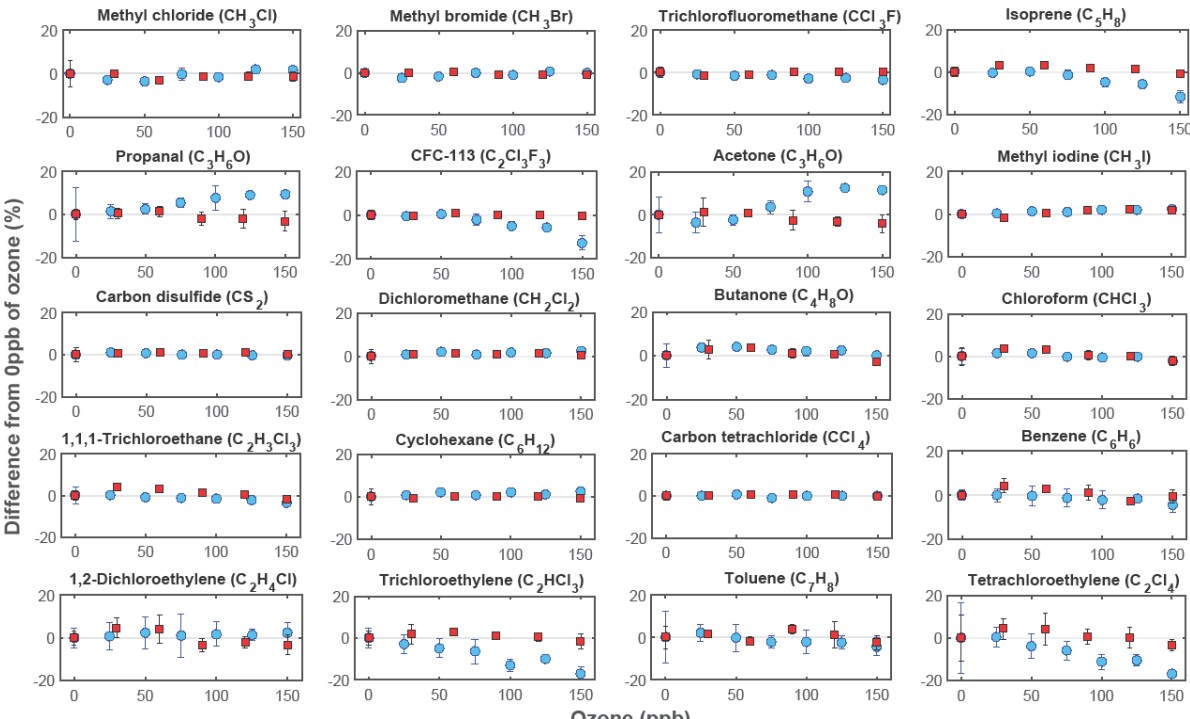

**Figure 7.** Ozone experiments. A stable VOC mixing ratio ($\approx 0.5$ ppb) was sampled under 0-150 ppb of $O_3$ with (red squares) and without (cyan circles) the use of ozone scrubber.

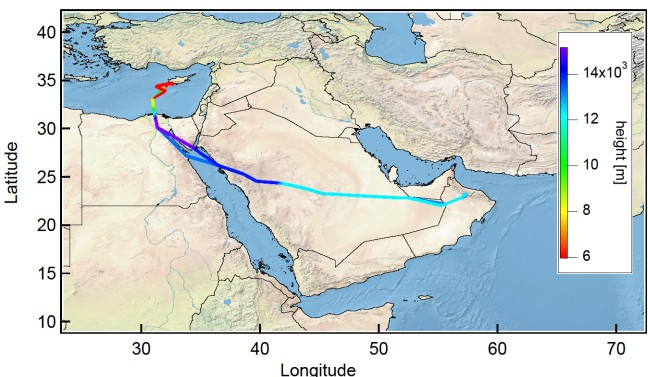

**Figure 8.** Route of research flight 20150813.

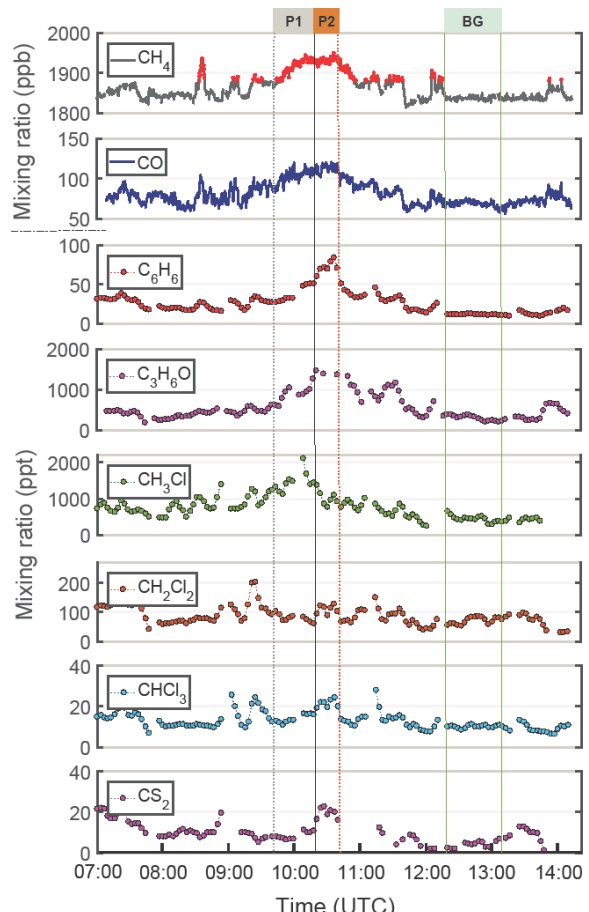

**Figure 9. Time series of eight compounds from flight 20150813.**

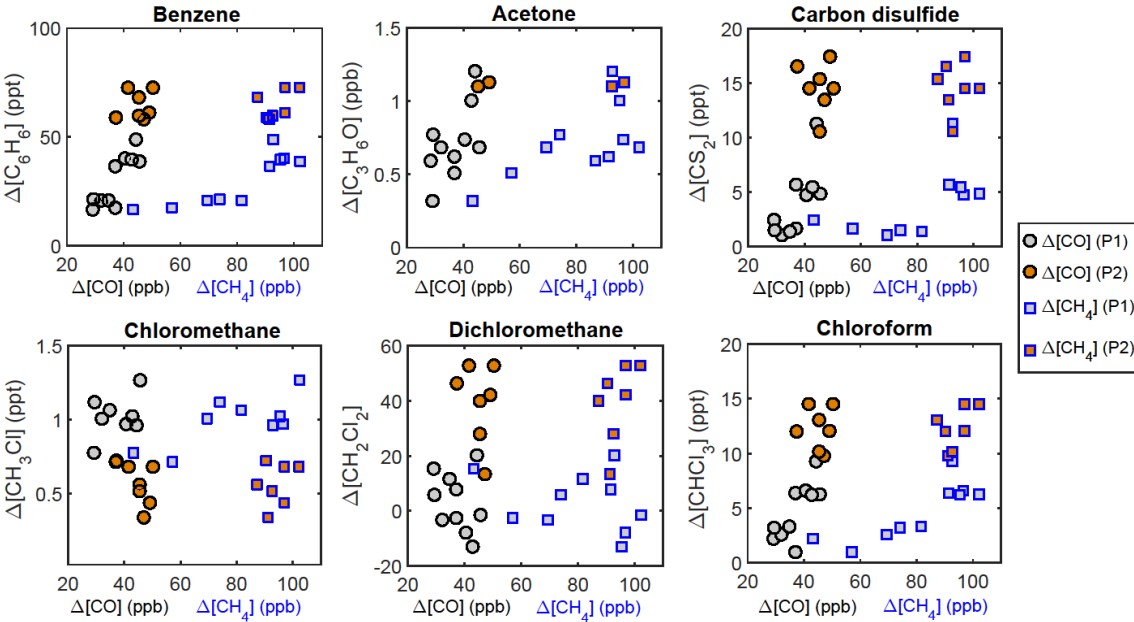

**Figure 10. Enhancement mixing ratio correlations in the two plumes.** Correlations of ExMR between CO (cycles) and CH₄ (squares) with benzene, acetone, carbon disulfide, chloromethane, dichloromethane and chloroform.

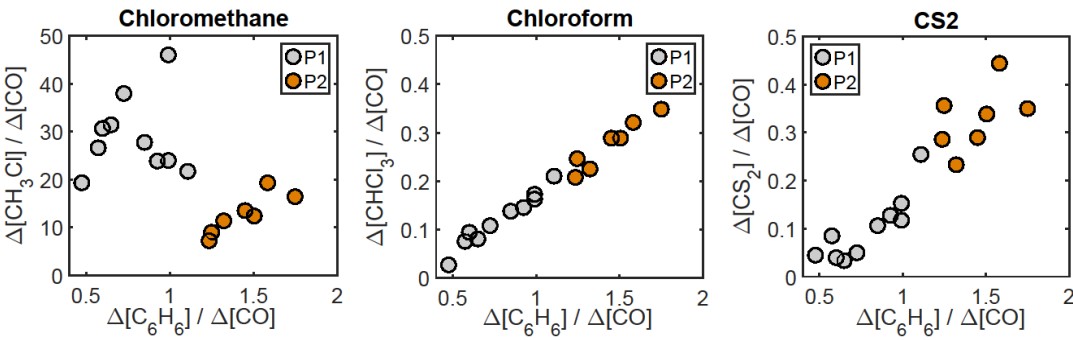

**Figure 11.** Comparison plots of NEMRs to CO for chloromethane, chloroform and carbon disulphide. The first part of the pollution plume P1 is indicated by grey bullets and the second part P2 with orange bullets.

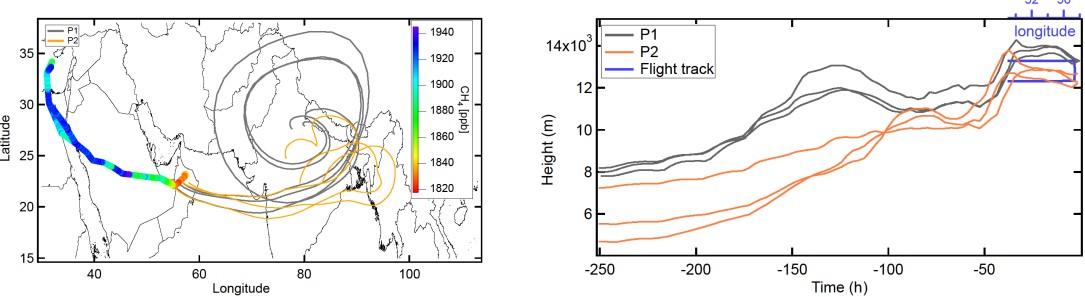

**Figure 12.** Back-trajectories for the plumes P1 and P2 according to geographic location (left) and in time (right).

35

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
