# Peer review of "An aircraft gas chromatograph-mass spectrometer System for Organic Fast Identification Analysis (SOFIA): design, performance and a case study of Asian monsoon pollution outflow"

_Atmospheric Measurement Techniques, 2017_

## Referee Comment (RC1) · Anonymous Referee #1 · 29 Aug 2017

Review of amt-2017-201 manuscript: "An aircraft gas chromatograph-mass spectrometer System for Organic Fast Identification Analysis (SOFIA): design, performance and a case study of Asian monsoon pollution outflow", by Efstratios Bourtsoukidis, Frank Helleis, Laura Tomsche, Horst Fischer, Rolf Hofmann, Jos Lelieveld, and Jonathan Williams

This article describes the technical design and first application of a fast preconcentration – gas chromatography/mass spectrometry (GC/MS) system for analysis of ambient organic trace gases from the HALO aircraft platform.

This manuscript is in general well written. The engineering of the instrument is impressive.  The system builds on a cryogenic preconcentration technique on an open tubular micro-trop with a second focusing step on a cooled GC column section.   GC separation is achieved on a commercial column that is contained in a low thermal mass and fast response heater.  Detection relies on mass spectrometry with an Agilent MSD.  These all are more or less established analytical principles and technical approaches that have been demonstrated in previous research and publications.  The most remarkable accomplishment is the fast time resolution that can be achieved with this system, allowing a full analytical cycle in 2-3 min.  But again, this has also been accomplished before, for instance with the TOGA instrument (see summary in Table 2).  Therefore, overall, I see relatively little novelty in this work.

For a paper describing a new analytical development, I would have liked to see a more in depth characterization of the analytical system through analytical testing.   The application examples unfortunately in my opinion do not serve that purpose.  Examples of such analytical experiments are for instance:

- Demonstrate neglect of water vapor effects by running test series at varying humidity.
- Demonstrate robustness against CO2 interference.  Under the chosen trapping conditions most of the CO2 is expected to be focused.  Upon desorption this will replace carrier gas and alter the chromatography.  That's why cryogenic focusing systems used by other groups often use CO2 removal agents.  It doesn't seem like this was tested here?
- Show linearity of the system.
- Compare response in synthetic versus whole air standards.
- Investigate interference from ozone.
- Demonstrate how detection limits were determined.

My most severe concern is the lack of addressing interferences from ozone in the sample air.  Given that this is an aircraft instrument with a high ceiling altitude (15 km, page 12/line 6), significantly higher (than surface) ozone levels will be regularly sampled.  Problems from co-collection of ozone have been recognized some 20+ years [*Goldan et al.*, 1995].  There are multiple effects that can occur, such as artifact formation from reaction of ozone in the sampling system, as well as loss of analytes during the prefocusing from oxidation by ozone.  Consequently, there is potential for positive and negative artifact effects. Ozone interference has been well documented in the peer-reviewed literature, see for instance [*Bates et al.*, 2000; *Plass-Duelmer et al.*, 2002; *Pollmann et al.*, 2005; *Lee et al.*, 2006; *Apel et al.*, 2008;

*Arnts*, 2008; *Hellen et al.*, 2012].   Modern analytical systems relying on VOC preconcentration systems followed by GC analysis take the ozone interferences into account, and mitigate this problem by selective removal of ozone in the sample air.   Analytical protocols and standard operating procedures developed through ACTRIS and by the WMO/GAW program (i.e. https://www.wmo.int/pages/prog/arep/gaw/documents/4thGAWVOC_DWD_ACTRIS-GAW_RecomRequire.pdf; https://www.wmo.int/pages/prog/arep/gaw/documents/5thGAW-VOC-Plass-Dulmer_OVOC.pdf; and WMO-GAW VOC Measurement Guidelines, in prep.) recommend use of ozone scrubbing techniques.

Given this plethora of information on this interference, and the obvious analytical biases that can occur, I think at this point in time enrichment/GC analytical systems need to consider ozone management for achieving good measurement quality.   I recommend against publishing analytical GC VOC analysis work that neglects addressing the important issue of ozone management.

Other comments:

Figure 1:

- I find this figure hard to read and recommend enlarging it.  Further, even after studying it for a while I had a hard time following the flow pathways.  I recommend adding in a supplement further versions of this figure that show highlighted with color the most important flow paths during important analytical sequence steps.
- For the zero/cal unit, shouldn't the zero supply gas be air, and shouldn't be the flow arrow go the other direction?
- How does the system get sufficient flow at the very low outside cabin pressures at 15 km altitude?  I don't fully understand the purpose of the 'Inlet FC' when there is another 'Sample FC' downstream in the path?

Page 3/Line 30: Define MuPo when first mentioned.

3/34-38: The figure is lacking this detail?

4/4:  Is the catalyst indeed Pt or platinum oxide?

4/10-12: Provide full detail on the standard that was used, i.e. components, compound mole fractions, and preparation date.

4/11:  …. 5 ml min$^{-1}$ ….

4/27:  Shouldn't one use a 'vacuum pump' rather than a 'sampling pump' for this application?

4/27:  NTC sensor is not shown in schematic.

4/31: Please provide pressure sensor specs.

5/17: …converter.

6/8-9: Please provide more detail on the pressure controller fabrication.

7/25-30: The plumbing diagram does not show how dilution series can be done with the system.

8/19: Please explain the 'in-flight pressure calibration curve'.

8/21: Please be more clear with the wording here. I recommend using the term 'precision error'. A low precision would actually reflect a high standard deviation, so a 'high' value.

11/12: I would not necessarily agree with this statement. In proper configuration the MSD is probably as fast of a GC detector as any other GC detection method.

11/37: Replacement of the carrier gas by $CO_2$ released into the flow path during sample desorption could be a major chromatographic interference.

Table 1: The precision result needs to be reported with the mole fraction at which it was determined.

Figure 4: A figure showing an ambient sample would be more valuable. Synthetic standards are usually easier to measure.

Figure 8: Mole fraction scale on y-axis in lower graph is missing.

**Reference cited:**

Apel, E. C., T. Brauers, R. Koppmann, B. Bandowe, J. Bossmeyer, C. Holzke, R. Tillmann, A. Wahner, R. Wegener, A. Brunner, M. Jocher, T. Ruuskanen, C. Spirig, D. Steigner, R. Steinbrecher, E. Gomez Alvarez, K. Muller, J. P. Burrows, G. Schade, S. J. Solomon, A. Ladstatter-Weissenmayer, P. Simmonds, D. Young, J. R. Hopkins, A. C. Lewis, G. Legreid, S. Reimann, A. Hansel, A. Wisthaler, R. S. Blake, A. M. Ellis, P. S. Monks, and K. P. Wyche (2008), Intercomparison of oxygenated volatile organic compound measurements at the SAPHIR atmosphere simulation chamber, *Journal of Geophysical Research-Atmospheres*, *113*(D20), doi:10.1029/2008jd009865.

Arnts, R. R. (2008), Reduction of Biogenic VOC Sampling Losses from Ozone via trans-2-Butene Addition, *Environmental Science & Technology*, *42*(20), 7663-7669, doi:10.1021/es800561j.

Bates, M. S., N. Gonzalez-Flesca, R. Sokhi, and V. Cocheo (2000), Atmospheric volatile organic compound monitoring. Ozone induced artefact formation, *Environmental Monitoring and Assessment*, *65*(1-2), 89-97, doi:10.1023/a:1006420412523.

Goldan, P. D., W. C. Kuster, F. C. Fehsenfeld, and S. A. Montzka (1995), Hydrocarbon measurements in the southeastern United States: The Rural Oxidants in the Southern Environment (ROSE) program 1990, *Journal of Geophysical Research-Atmospheres*, *100*(D12), 25945-25963, doi:10.1029/95jd02607.

Hellen, H., P. Kuronen, and H. Hakola (2012), Heated stainless steel tube for ozone removal in the ambient air measurements of mono- and sesquiterpenes, *Atmospheric Environment*, *57*, 35-40, doi:10.1016/j.atmosenv.2012.04.019.

Lee, J. H., S. A. Batterman, C. R. Jia, and S. Chernyak (2006), Ozone artifacts and carbonyl measurements using Tenex GR, Tenex TA, Carboparck B, and Carbopack X adsorbents, *Journal of the Air & Waste Management Association*, *56*(11), 1503-1517.

Plass-Duelmer, C., K. Michl, R. Ruf, and H. Berresheim (2002), C-2-C-8 hydrocarbon measurement and quality control procedures at the Global Atmosphere Watch Observatory Hohenpeissenberg, *Journal of Chromatography A*, *953*(1-2), 175-197, doi:10.1016/s0021-9673(02)00128-0.

Pollmann, J., J. Ortega, and D. Helmig (2005), Analysis of atmospheric sesquiterpenes: Sampling losses and mitigation of ozone interferences, *Environmental Science & Technology*, *39*(24), 9620-9629.

---

## Referee Comment (RC2) · Anonymous Referee #2 · 4 Sep 2017

Review of amt-2017-201 manuscript: "An aircraft gas chromatograph-mass spectrometer System for Organic Fast Identification Analysis (SOFIA): design, performance and a case study of Asian monsoon pollution outflow", by Efstratios Bourtsoukidis, Frank Helleis, Laura Tomsche, Horst Fischer, Rolf Hofmann, Jos Lelieveld, and Jonathan Williams

This manuscript is well written and describes an aircraft based pre-concentration GC-

[Figure]

MS for identification of volatile organics in-situ. The manuscript references the TOGA instrument and shows similarities with this instrument but includes some very interesting innovations such as the LN2 implementation and the oven design. Unfortunately, it is the detailed description of these two elements that seems most obviously lacking from the text. Firstly, there seems to be a problem with the overall order of the manuscript. The sampling unit is described with reference to the liquid nitrogen container and is particularly difficult to understand without reading later sections and then coming back to re-read the descriptions and try to build a picture of how the instrument works. The figures include CAD drawings and a photograph but the picture quality is much too small to make out any useful detail. I'm still a little unsure as to how the cooling system actually operates.

I have read the comments from the other reviewer before writing this review and I do agree with many of their observations. However, I do feel that it is vitally important to publish these types of manuscript, which describe in detail the operation of an instrument that will undoubtedly be used in many future research campaigns. Fitment of this type of instrument in any aircraft is a large undertaking and cannot be held in the same regard as say, a laboratory instrument. To this respect, even if the instrument as a whole is not novel (although I would argue that this indeed is), there are many novel design aspects and operational challenges that should be reported in this type of manuscript and are extremely useful information for the scientific community.

I do agree that the manuscript should focus more on performance testing of the instrument and feel that the campaign data section adds very little to the overall text. I would like to see more detail, text and graphic, describing the instrument. It is currently very difficult to understand the system in its entirety. If a manuscript is to describe an instrument then the relevant detail should be present to allow the reader to fully understand the system.

Observations from the text:

All units should have a space after the value- e.g. 2 mL not 2mL

P1,L24 – ExR should maybe be ExMR with the reference to mixing ratio matching the acronym NEMR

P2,L27 – I believe the UK aircraft operated by FAAM has a GCMS on-board. I think the first reference to this is from the Co-ordinated Airborne Studies in the Tropics (CAST) campaign.

L32 – include quotation - "System for Organic Fast Identification Analysis"

P3, L30 - In my experience, heated teflon line can be quite permeable, how do you test for contamination? e.g. TOGA has zero air introduced right at the start of the inlet as it enters the cabin.

Section 2.1.1 – I'm not sure that I fully understand the inlet system with regard to the pumps. Maybe clarify on the schematic. Which pumps are where? inlet pump or sample pump? also, not clear how the calibrated volume is used. Is it pressurised using the pump and then the pressure supplies the trap? Need to make this more clear.

P3, L30 - brings the air into the inlet MuPo

L36 - so the KNF pumps used were all-PFA? maybe change the sentence around to say 3 identical PFA pumps were used....the Pmax of single stage wasn't enough, dual stage too heavy, so backed with 4th pump...

P4, L1 - what are chronometer o-rings? watch o-rings? reference part number, material type? how were they fitted? did you have to machine a groove? Also, did you test the pumps for inertness?

P4, L11 - 5mL/min?? volumetric or mass flow? SCCM better?

P4, L15- strange use of the word concert in a scientific context. Maybe 'concurrently' or 'in unison' would be better?

P5, L3 – aerogel product number, ref? L4 -3D printed material

P5 L12- stainless tubing but what grade? Any coating? Is this the trap material in contact with the sample? Text quite ambiguous

P5, L17 - how are the traps electrically isolated from the fittings and thermocouple? fitting and ferrule material?

P6, L11 - is the CPOR the name for the system built using the camping gas regs or is this a separate commercial item? Unclear

P6, L32 - this makes it sound like the system is able to run at sub-ambient? is this possible as it seems that the oven is cooled by a fan

P7, L11 – to which regulator are you referring? temperature controllers? are you saying that PID values are variable dependent upon temp or do you refer to the combination of fan speed and heater control. paragraph needs clarification

P7, L25 – 'The dwell time for the individual ions selected was 10ms and a complete chromatogram run for 2.4min' Bad sentence structure, maybe: . . . and the chromato­graphic runtime was 2.4 min P7, L30 - Apel Riemer standard gas referenced to any scales? date of cylinder production?

P7, L36 - creates the need for robust peak

P7, L38 - separation of peaks that elute at similar retention times

L40 - peaks. MPIC-Chrom, a new peak integration software written in IGOR, was used. . .

P8, L11 – too many words, maybe simplify: 50 degC initial temp, hold for 20 sec, ramp to 80 degC at 2 degC/sec, ramp to 150 at 1. . ..

P8, L15 - if you are going to mention this then need to detail the improvements

P10, L39 – 'were' should maybe be 'where'

---

## Referee Comment (RC3) · Anonymous Referee #3 · 11 Sep 2017

Summary:

Fast GC-MS measurements for the in situ analysis of VOCs represent an important analytical tool in understanding complex and often rapidly-changing air mass compositions, especially aboard mobile platforms such as aircraft. The authors describe a newly constructed in-situ GC-MS system that is deployable on the High Altitude and Long Range Research Aircraft (HALO). While many of the underlying principles for the

[Figure]

SOFIA system have been in use in various GC-MS systems in one form or another, the design of the individual components of SOFIA is new and offers an additional resource for instrument engineers. For example, the sample conditioning and pre-concentration components are a new design capable of achieving a range of cold temperatures for three separate traps utilizing a single liquid nitrogen reservoir that does not need to be refilled for up to 17 hours. The manuscript is concise, well written, and will benefit the scientific community. I recommend publication after the reviewer comments are addressed.

General comments:

- I would like to see a brief discussion on the power requirements and pre/post-flight protocols. These are all critical considerations for a flight-based instrument. How much power does the entire system draw? How much time does it take to get the instrument online (heaters up, reservoirs cold, MSD's pumped down, etc.)? Also, the information in the last paragraph of the discussion section (size and weight) should be moved to Section 2 along with the above discussion.

- There needs to be a discussion on ozone artifacts and/or mitigation, particularly since this instrument is designed for high altitude sampling where high levels of strato-spheric ozone may be encountered and given the fact that the authors report isoprene, a species that is reactive with ozone.

- Since this manuscript details the instrument performance, I would like to see more discussion on the calibrations. The author state that the response (of just the detector or the whole sampling system?) is linear for mixing ratios ranging from few ppt up to 3 ppb (Section 2.4). That is impressive for an open tubular trap that does not contain a substrate such as glass beads that increase the surface area for absorption thereby minimizing nonlinear surface effects. How were the calibrations conducted (e.g., dynamic or static dilutions)? For dynamic dilutions, did that include air from the catalyst and therefore ambient levels of water or was it a dry sample stream? How does the

in-flight calibration dynamic range compare to the observations? How is LOD determined? What about breakthrough at higher VOC concentrations? What range of VOCs to the author's realistically expect to be able to measure such highly volatile compounds (e.g., ethane, ethene, ethyne), high molecular weight species (e.g., C9 aromatics), or highly polar species (e.g., butanal)? What are the current sampling limitations of this instrument?

Technical comments:

- The abstract should include a description of the analytical range of the instrument including the types of compounds that can be measured.

- MuPo is not defined until page 4 (P4), but first appears on P3. I suggest getting rid of the "MuPo" nomenclature entirely as it is unnecessary jargon – it's a valve. The vast majority of GC-MS systems have these types of valves and I have never heard it called a "MuPo."

- P4L15: It should be stated clearly that the ambient air is also a possible gaseous input to the sample selector valve.

- P4L28: Is the calibrated volume evacuated prior to sampling? It's not clear if the sample is pushed or pulled through the cold traps. Is the calibrated volume evacuated and therefore acts to pull the sample through the traps since the sample pump is isolated from the sample stream during sampling? Or, does the inlet pump create enough pressure to push the sample through the traps and pressurize the calibrated volume to a set pressure?

- P11L35: The NOAA GC-MS system (Lerner et al., AMT 2017) also utilizes a Stirling cooler. Depending on the cooler's lift capacity, Stirling coolers can easily obtain minimum temperatures below that of liquid nitrogen (i.e., -200 °C). The cooling rates obtained for the NOAA GC-MS trapping system are in the range of – 6 to -10 °C/second for a larger, dual-channel system.

- Figure 1 (F1) is hard to read and decipher, partly because the resolution is too low. Additional corrections/suggestions for Figure 1 are below:

F1a.) I am confused about the purpose of the 6-port inlet valve. No matter what position the valve is in, the Zero/Calibration unit will always be in the sample stream, never bypassed. The only thing that changes is whether the sample pump is upstream or downstream of the Zero/Calibration unit. I'm assuming this has something to do with managing the hot catalyst exhaust, but it's not clear in the diagram or text.

F1b.) Is there a four-way union directly downstream of the inlet valve and upstream of the catalyst? If so, wouldn't the catalyst represent a large dead volume as it is always exposed to the sample stream?

F1c.) Why is there an arrow to the right of the N2? Shouldn't the flow be going the other way to flush the catalyst when not exposed to ambient air? If so, this is not described in Section 2.1.1.

F1d.) The authors missed a few of the sample line crossover junctions (e.g., just below the sample selector valve and just below HALO exterior).

F1e.) I would suggest overlaying the arrows indicating flow direction over the various connections on the valve itself as well as clearly stating which position the valve is in (i.e., inject or transfer).

F1f.) There should also be a key to clearly identify 2-way, 3-way, needle valves, etc.

- Figure 2. It is difficult to know what each part is without added annotations.

---

## Referee Comment (RC4) · Anonymous Referee #4 · 15 Sep 2017

The manuscript by Bourtsoukidis et al. describes a new in-situ fast GC/MS system. The instrument adds interesting new engineering developments and is a useful addition to the suite of instruments now available for airborne trace organic gas analysis. I agree with the other reviewers comments and don't want to repeat too much here. The most essential improvements would be 1) focus more detail on the optimization and testing done to determine performance; 2) reduce the discussion of the Asian monsoon outflow (a time series of trace gases would be sufficient to demonstrate the

performance in actual flight research mode); and 3) improve the clarity of the figures (the CAD drawings don't really help understand the functional design of components; the flow diagram needs to be clearer).

Beyond these major revisions and those comments of the other reviewers, I had a few other questions/comments:

P3, L29: The inlet needs a complete description. The reference by Wendisch has no detail on the design, testing, and use of the inlet for trace gas analysis. This is a critical part of the system, and as far as I can tell from the flow diagram, calibration and zero additions are done downstream of the inlet. Thus, the testing that was done (or not) to determine inlet effects under ambient type conditions is important for the performance of the overall instrument.

2.1.1. Cal/Zero control. Would be interested to know if the lines with zero/cal gases are flushed continuously. Intermittent sampling of a calibration tank could cause artifacts.

2.2.3 Liquid nitrogen container description might be better discussed prior to section 2.2.2 on Trap temperature control.

2.2.4. This section on pressure control is not really clear at all.

P7 L25. Were different dwell times tested, or was the effect of total dwell time of multiple ions tested? Basically, it would be important to know exactly how these parameters impacted both signal to noise (from dwell time) and precision/accuracy (from measurements across the chromatographic peak).

Other questions:

Ambient water variations: Often inlets demonstrate artifacts when the aircraft crosses from very dry to very moist conditions. I'd be interested to see examples of instrument performance (esp. for OVOC) under alternating conditions of wet and dry air. Calibration method is unclear. Could this be described in more detail, and also describe an actual sequence of sample/zero/sample/calibration/sample etc. that is used during

research flights.

P9 L 10. Measured variation of CCl4 is slightly more than 10%, and the bulk of this variation should be from the instrument, not the ambient CCl4. A precision of 10% seems high. Is this typical?

Table 1. Could you explain the value of 5% for uncertainty in sample volume? This seems too high given the quality of pressure and temperature measurements.

[Figure]

---

## Author Comment (AC1) · 13 Oct 2017

**Reviewer #1**

This article describes the technical design and first application of a fast preconcentration – gas chromatography/mass spectrometry (GC/MS) system for analysis of ambient organic trace gases from the HALO aircraft platform.

This manuscript is in general well written. The engineering of the instrument is impressive. The system builds on a cryogenic preconcentration technique on an open tubular micro-trop with a second focusing step on a cooled GC column section. GC separation is achieved on a commercial column that is contained in a low thermal mass and fast response heater. Detection relies on mass spectrometry with an Agilent MSD. These all are more or less established analytical principles and technical approaches that have been demonstrated in previous research and publications. The most remarkable accomplishment is the fast time resolution that can be achieved with this system, allowing a full analytical cycle in 2-3 min. But again, this has also been accomplished before, for instance with the TOGA instrument (see summary in Table 2). Therefore, overall, I see relatively little novelty in this work.

We thank the reviewer for acknowledging the engineering advances of our instrument. While our system has indeed some similarities with TOGA (and indeed all such GC systems), the cryogenic pre-concentrator, the GC oven, the design and electronics control are markedly different and novel. Our approach to use differential pressure of LN2 container to flood the traps is a unique approach which allows considerable reduction in the size compared to TOGA and this is substantially different than the approach used in the rest of fast GC instruments described in Table 2. We believe that we have elaborately discussed all differences, advantages and disadvantage of our instrument. Given the fact that there are only 4 online fast GC instruments that are used for airborne research we believe that there are many novel aspects in our paper (as recognized by the other 3 reviewers).

For a paper describing a new analytical development, I would have liked to see a more in depth characterization of the analytical system through analytical testing. The application examples unfortunately in my opinion do not serve that purpose. Examples of such analytical experiments are for instance:

- Demonstrate neglect of water vapor effects by running test series at varying humidity.

- Demonstrate robustness against CO2 interference. Under the chosen trapping conditions most of the CO2 is expected to be focused. Upon desorption this will replace carrier gas and alter the chromatography. That's why cryogenic focusing systems used by other groups often use CO2 removal agents. It doesn't seem like this was tested here?

- Show linearity of the system.

- Compare response in synthetic versus whole air standards.

- Investigate interference from ozone.

- Demonstrate how detection limits were determined.

My most severe concern is the lack of addressing interferences from ozone in the sample air. Given that this is an aircraft instrument with a high ceiling altitude (15 km, page 12/line 6), significantly higher (than surface) ozone levels will be regularly sampled. Problems from co-collection of ozone have been recognized some 20+ years [Goldan et al., 1995]. There are

multiple effects that can occur, such as artifact formation from reaction of ozone in the sampling system, as well as loss of analytes during the prefocusing from oxidation by ozone. Consequently, there is potential for positive and negative artifact effects. Ozone interference has been well documented in the peer-reviewed literature, see for instance [Bates et al., 2000; Plass-Duelmer et al., 2002; Pollmann et al., 2005; Lee et al., 2006; Apel et al., 2008; Arnts, 2008; Hellen et al., 2012]. Modern analytical systems relying on VOC preconcentration systems followed by GC analysis take the ozone interferences into account, and mitigate this problem by selective removal of ozone in the sample air. Analytical protocols and standard operating procedures developed through ACTRIS and by the WMO/GAW program (i.e. https://www.wmo.int/pages/prog/arep/gaw/documents/4thGAWVOC_DWD_ACTRIS-GAW_RecomRequire.pdf; https://www.wmo.int/pages/prog/arep/gaw/documents/5thGAW-VOC-Plass-Dulmer_OVOC.pdf; and WMO-GAW VOC Measurement Guidelines, in prep.) recommend use of ozone scrubbing techniques.

Given this plethora of information on this interference, and the obvious analytical biases that can occur, I think at this point in time enrichment/GC analytical systems need to consider ozone management for achieving good measurement quality. I recommend against publishing analytical GC VOC analysis work that neglects addressing the important issue of ozone management.

We agree with the reviewer that the aforementioned experiments would make this instrumental description paper more comprehensive. In the revised version we have addressed these issues where appropriate, and have now included more descriptive text, experiments, figures and CAD drawings in order to better present the characteristics of our instrument. Special attention was given to the ozone interference as it was emphasized as the most critical issue by the reviewer. Our additions are summarized as following:

- Water vapor: The effect of water vapor was investigated through the detection efficiency of the specified compounds of interest. Calibration gas was sampled under the extreme conditions dry (0% RH) and wet (100% RH). It was found that the detection efficiency was not altered for any of the stated species. These results are now integrated into a new figure on which we illustrate both dry and humid calibrations up to 3ppb.
- $CO_2$ interference: As mentioned the $CO_2$ interference does not impact the investigated species. First, the calibrations are performed with ambient $CO_2$ levels and therefore substantial changes in the chromatography between ambient and calibration measurements are not expected. Second, the ions that are monitored and presented in this study so not interfere with the respective ions of $CO_2$. As recommended in the suggested by the reviewer ARCTIS and WMO/GAW program, no action of $CO_2$ removal should be taken but a characterization of the effect. Nonetheless, we now include a more extended discussion on the limitations that are induced due to the fact that $CO_2$ is trapped in the system.
- Linearity of the system: We initially considered it as unnecessary to show (rather than state) the linearity of a mass spectrometer that has been already published in a plethora of studies. However, the reviewers are correct that it's not only the detector responsible for the linearity but the whole system itself. Therefore, we have included a new figure (supplementary information Fig. S2) that displays the linear calibration curves for all species mentioned in Table 2.
- Compare response in synthetic versus whole air standards: Since we have now demonstrated the effect of moisture and we present chromatograms from both

calibration gas and ambient samples, the detector response via comparison of synthetic versus whole air standards has also been covered.

- Investigate the interference of ozone: In the revised version we have included a new section in which we fully characterize the effect of ozone with laboratory experiments.
- Demonstrate how detection limits were determined. The following sentence has been added into the text ''*The detection limits were determined as three times the standard deviation of the signal produced by 10 zero air samples in three different concentration levels (i.e. ≈125ppt, ≈250ppt, ≈500ppt).*''

Other comments:

Figure 1:

- I find this figure hard to read and recommend enlarging it. Further, even after studying it for a while I had a hard time following the flow pathways. I recommend adding in a supplement further versions of this figure that show highlighted with color the most important flow paths during important analytical sequence steps.
Following the advice by the reviewer, we have now enlarged the figure and include a supplement on which we illustrate the flow paths during the most important analytical sequence steps.

- For the zero/cal unit, shouldn't the zero supply gas be air, and shouldn't be the flow arrow go the other direction?
We hope that the supplementary information figures are now clarifying the flow paths followed during the zero supply gas.

- How does the system get sufficient flow at the very low outside cabin pressures at 15 km altitude? I don't fully understand the purpose of the 'Inlet FC' when there is another 'Sample FC' downstream in the path?
We now include the following sentence: "*The inlet pump and the inlet FC pressurize the sample to enhance the throughput in the inlet tubes (200sccm) over the actual sampling flow (40sccm). This allows faster conditioning of the inlet line (see Fig. S1)*"

- Page 3/Line 30: Define MuPo when first mentioned
The term MuPo is now removed from the manuscript.

3/34-38: The figure is lacking this detail?
This detail is indeed lacking from the figure that however indicates it as a single inlet pump. To avoid similar misconceptions the text in now revised as: "*Therefore, three small identical PFA pumps (NMP 850KNDC, KNF Global strategies AG) were run used on the low pressure inlet in parallel and since the maximum pressure of the single stage was not enough, it was backed by a fourth pump connected to the high pressure exit upstream of the sampling system (all four pumps are illustrated as a single one in Fig. 1).*"

4/4: Is the catalyst indeed Pt or platinum oxide?
The catalyst is platinum on alumina (PN 206016, Sigma Aldrich, USA) and this is now stated in the text.

4/10-12: Provide full detail on the standard that was used, i.e. components, compound mole fractions, and preparation date.
*The information is added but without the preparation date as several calibration gas mixtures have been used over the development of the instrument. Instead we note that "The calibration standards used were within the manufacturers guaranteed accuracy period of two years."*

4/11: .... 5 ml min-1....
*Corrected to 5sccm*

4/27: Shouldn't one use a 'vacuum pump' rather than a 'sampling pump' for this application?
*Sample pump is a vacuum pump, because it is after the sampling point. It is now indicated in the text inside brackets "...a sample diaphragm pump (vacuum pump; Pfeiffer MVP 006-4) and a NTC temperature sensor (Y3k-type thermocouple)"*

4/27: NTC sensor is not shown in schematic.
*Temperature sensors are in almost every part of the instrument and including all of them in the schematic would have made it more difficult to read.*

4/31: Please provide pressure sensor specs.
*We added the specifications. In general, NTC sensors are type Y3k for temperatures close to ambient and k-type thermocouples for all high temperature heaters.*

5/17: ...converter.
*Thank you for the correction.*

6/8-9: Please provide more detail on the pressure controller fabrication.
*We have now included additional text and a schematic.*

7/25-30: The plumbing diagram does not show how dilution series can be done with the system.
*Dilutions are performed with variable calibration gas flows. We have now included supplementary information over which we are illustrating the flow paths of the calibration.*

8/19: Please explain the 'in-flight pressure calibration curve'.
*The text was modified as following: "In order to assess possible pressure dependencies, calibrations were performed at each pressure level during the flight."*

8/21: Please be more clear with the wording here. I recommend using the term 'precision error'. A low precision would actually reflect a high standard deviation, so a 'high' value.
*Following the recommendation by the reviewer we have revised this sentence accordingly.*

11/12: I would not necessarily agree with this statement. In proper configuration the MSD is probably as fast of a GC detector as any other GC detection method.
*Our statement is not about speed but on the restrictions that arise when a large number of species are monitored inside the same SIM window. If many species are monitored simultaneously then the resolution of each peak will be reduced and only few points will construct the peak. Generally we aimed to define the peaks with at least 6 points. Remember this MS is a quadrupole which must sequentially analyze the ions., not a high frequency ToF. Hence, it is still a limitation and this is the information we wanted to convey with that sentence.*

11/37:  Replacement of the carrier gas by CO2 released into the flow path during sample desorption could be a major chromatographic interference.

*We hope that we have sufficiently addressed this comment with our additional discussion. "In any case, low adsorption temperatures will result in trapping CO2 which can induce chromatographic and detection problems depending on the selected ions monitored. In our system the most abundant atmospheric gases (nitrogen, oxygen, argon) will be mostly removed from the sample, but the less volatile gases such as CO2 are trapped. The elution of CO2 restricts the range of the analytes that can be monitored (e.g. acetaldehyde). Since the species monitored Nonetheless, the selected species that are implemented in our method do not have interferences with the parent CO2 ions and hence, our measurements were not influenced by ambient CO2."*

Table 1: The precision result needs to be reported with the mole fraction at which it was determined.

The precision values that are reported in the table are the result of three different mole fractions (3 levels), similar to the detection limits that have been described above. This information is now added in the description of Table 1.

Figure 4:  A figure showing an ambient sample would be more valuable.  Synthetic standards are usually easier to measure.

The ambient samples that have been obtained during the OMO campaign were monitoring only half of the species that are now implemented in our method. Therefore we would suggest keeping Fig. 4 with laboratory calibrations as it depicts better the capabilities of our system comparing with the initial configuration (see P8L2-3 of the ACPD manuscript). In addition, we now include a supplementary figure on which we illustrate the chromatogram of an ambient sample that was taken during the case study flight.

Figure 8:  Mole fraction scale on y-axis in lower graph is missing.

Thank you for pointing this out. Given the comments received by the other reviewers Fig. 8 is now removed from the manuscript.

**Reviewer #2**

This manuscript is well written and describes an aircraft based pre-concentration GC- MS for identification of volatile organics in-situ.  The manuscript references the TOGA instrument and shows similarities with this instrument but includes some very interesting innovations such as the LN2 implementation and the oven design.  Unfortunately, it is the detailed description of these two elements that seems most obviously lacking from the text.  Firstly, there seems to be a problem with the overall order of the manuscript.  The sampling unit is described with reference to the liquid nitrogen container and is particularly difficult to understand without reading later sections and then coming back to re-read the descriptions and try to build a picture of how the instrument works.  The figures include CAD drawings and a photograph but the picture quality is much too small to make out any useful detail.  I'm still a little unsure as to how the cooling system actually operates.

I have read the comments from the other reviewer before writing this review and I do agree with many of their observations. However, I do feel that it is vitally important to publish these types of manuscript, which describe in detail the operation of an instrument that will undoubtedly be used in many future research campaigns. Fitment of this type of instrument in any aircraft is a large undertaking and cannot be held in the same regard as say, a laboratory instrument. To this respect, even if the instrument as a whole is not novel (although I would argue that this indeed is), there are many novel design aspects and operational challenges that should be reported in this type of manuscript and are extremely useful information for the scientific community.

I do agree that the manuscript should focus more on performance testing of the instrument and feel that the campaign data section adds very little to the overall text. I would like to see more detail, text and graphic, describing the instrument. It is currently very difficult to understand the system in its entirety. If a manuscript is to describe an instrument then the relevant detail should be present to allow the reader to fully understand the system

We highly appreciate the comments and feedback from the reviewer. In the revised version we have thoroughly addressed the main comments made above (i.e. CAD drawings, extended description of the LN2 system and oven, order of the manuscript).

Observations from the text:

All units should have a space after the value- e.g. 2 mL not 2mL
Thank you for the suggestion. All units were modified accordingly.

P1,L24 – ExR should maybe be ExMR with the reference to mixing ratio matching the acronym NEMR
Thank you for the suggestion. The acronym is now ExMR.

P2,L27 – I believe the UK aircraft operated by FAAM has a GCMS on-board. I think the first reference to this is from the Co-ordinated Airborne Studies in the Tropics (CAST) campaign.
According to N. R. P. Harrrris et al. (2017) (https://doi.org/10.1175/BAMS-D-14-00290.1) sampling of organohalogens was performed with Whole Air Samples (WAS) while the already sited fast GC (Gostlow et al., 2010) was operated in the ground during the campaign.

L32 – include quotation - "System for Organic Fast Identification Analysis"
It is now included.

P3, L30 - In my experience, heated teflon line can be quite permeable, how do you test for contamination? e.g. TOGA has zero air introduced right at the start of the inlet as it enters the cabin.
The contamination was tested in a similar manner. On board SOFIA a N2 bottle (6.0) was frequently used for diagnostic purposes. This information has been now in the text.

Section 2.1.1 – I'm not sure that I fully understand the inlet system with regard to the pumps. Maybe clarify on the schematic. Which pumps are where? inlet pump or sample pump? also, not clear how the calibrated volume is used. Is it pressurized using the pump and then the pressure supplies the trap? Need to make this more clear.

We believe that the flow path schematics that are now included as supplementary information provide enough information for the reader to clearly understand how the pumps and calibrated volume are used during sampling. Additional text is also added to section 2.1.1.

P3, L30 - brings the air into the inlet MuPo
Corrected.

L36 - so the KNF pumps used were all-PFA? maybe change the sentence around to say 3 identical PFA pumps were used....the Pmax of single stage wasn't enough, dual stage too heavy, so backed with 4th pump...
Thank you for the suggestion. The text was modified accordingly.

P4, L1 - what are chronometer o-rings? watch o-rings? reference part number, material type? how were they fitted? did you have to machine a groove? Also, did you test the pumps for inertness?
They are watch o-rings that are typically from NBR (Nitrile Butadiene Rubber). They were fitted outside of the original Teflon diaphragm sealing edge of the pump heads, don't present any surface to the sample, serve only for He tightness. Inertness can be easily checked at pressures higher than ~600mbar by switching Inlet MuPo to the opposite position, placing pump and inlet flow controller after the sampling point. The text was modified accordingly: *"The o-rings were fitted outside the original Teflon diaphragm sealing edge of the pump heads and therefore they are not in contact with the sample."*

P4, L11 - 5mL/min?? volumetric or mass flow? SCCM better?
Changed to sccm.

P4, L15- strange use of the word concert in a scientific context. Maybe 'concurrently'or 'in unison' would be better?
Changed to concurrently.

P5, L3 – aerogel product number, ref? L4 -3D printed material
"Silica granules, InnoDämm, Germany" was added. It is actually a 3D printed elastomer enclosure.

P5 L12- stainless tubing but what grade? Any coating? Is this the trap material in contact with the sample? Text quite ambiguous
This information is now added in the text as *"The three traps are made of straight thin walled and uncoated stainless steel tubing (type 1.4301, Günther Lämmermeir OHG, Germany)"*

P5, L17 - how are the traps electrically isolated from the fittings and thermocouple? fitting and ferrule material?
This information is now added in the text: *"The traps were electrically isolated with stainless steel fittings and Teflon ferrules."*

P6, L11 - is the CPOR the name for the system built using the camping gas regs or is this a separate commercial item? Unclear
It is a separate item and custom build are mentioned in P6L8. We now include a schematic which illustrates how the pressure regulation works.

P6, L32 - this makes it sound like the system is able to run at sub-ambient? is this possible as it seems that the oven is cooled by a fan

The system cannot run in sub-ambient temperatures exactly because the oven is cooled by a fan. To avoid similar misunderstandings by other readers the sentence was modified as "*With the limited cooling power available for the GC oven, and to avoid condensation issues, the chromatography was run above ambient temperatures as the fan cannot create sub-ambient temperatures.*"

P7, L11 – to which regulator are you referring? temperature controllers? are you saying that PID values are variable dependent upon temp or do you refer to the combination of fan speed and heater control. paragraph needs clarification

We now clarify that we are referring to temperature regulator and the electronic regulation. We refer to a combination.

P7, L25 – 'The dwell time for the individual ions selected was 10ms and a complete chromatogram run for 2.4min' Bad sentence structure, maybe:...and the chromatographic runtime was 2.4 min.

Thank you for the suggestion. The text is modified as suggested.

P7, L30 - Apel Riemer standard gas referenced to any scales? date of cylinder production?

Changed to *using a multicomponent (79 species) calibration standard with mixing ratios of about 50 ppb (Apel-Riemer Environmental Inc.). The calibration standards used were within the manufacturers guaranteed accuracy period of two years.* "

P7, L36 - creates the need for robust peak

Corrected.

P7, L38 - separation of peaks that elute at similar retention times

Corrected.

L40 - peaks. MPIC-Chrom, a new peak integration software written in IGOR, was used…

Corrected.

P8, L11 – too many words, maybe simplify: 50 degC initial temp, hold for 20 sec, ramp to 80 degC at 2 degC/sec, ramp to 150 at 1...

The text was revised following the suggestion by the reviewer.

P8, L15 - if you are going to mention this then need to detail the improvements

The improvements are the fan control, being able to run below commercial starting rpm and the starting temperature was 60C. Oven heater has to be repaired. These improvements are now mentioned in the text.

P10, L39 – 'were' should maybe be 'where'

Thank you for the correction. We highly appreciate all the detailed feedback and linguistic suggestion provided by your side.

**Reviewer #3**

Fast GC-MS measurements for the in situ analysis of VOCs represent an important analytical tool in understanding complex and often rapidly-changing air mass compositions, especially aboard mobile platforms such as aircraft. The authors describe a newly constructed in-situ GC-MS system that is deployable on the High Altitude and Long Range Research Aircraft (HALO). While many of the underlying principles for the SOFIA system have been in use in various GC-MS systems in one form or another, the design of the individual components of SOFIA is new and offers an additional resource for instrument engineers. For example, the sample conditioning and pre-concentration components are a new design capable of achieving a range of cold temperatures for three separate traps utilizing a single liquid nitrogen reservoir that does not need to be refilled for up to 17 hours. The manuscript is concise, well written, and will benefit the scientific community. I recommend publication after the reviewer comments are addressed.

We thank the reviewer for acknowledging the novel aspects of our fast GC-MS system and we hope that we are sufficiently responding to all comments made.

General comments:
- I would like to see a brief discussion on the power requirements and pre/post-flight protocols. These are all critical considerations for a flight-based instrument. How much power does the entire system draw? How much time does it take to get the instrument online (heaters up, reservoirs cold, MSD's pumped down, etc.)? Also, the information in the last paragraph of the discussion section (size and weight) should be moved to Section 2 along with the above discussion.
We now include a discussion in the "2.7 Specifications during OMO campaign" chapter over which we are addressing all the above comments.

- There needs to be a discussion on ozone artifacts and/or mitigation, particularly since this instrument is designed for high altitude sampling where high levels of stratospheric ozone may be encountered and given the fact that the authors report isoprene, a species that is reactive with ozone.
We have added a new section on which we are thoroughly discussing the effects of ozone. We now illustrate the experiments performed in order to characterize the system under a wide range of ozone concentrations.

- Since this manuscript details the instrument performance, I would like to see more discussion on the calibrations. The author state that the response (of just the detector or the whole sampling system?) is linear for mixing ratios ranging from few ppt up to 3 ppb (Section 2.4). That is impressive for an open tubular trap that does not contain a substrate such as glass beads that increase the surface area for absorption thereby minimizing nonlinear surface effects. How were the calibrations conducted (e.g., dynamic or static dilutions)? For dynamic dilutions, did that include air from the catalyst and therefore ambient levels of water or was it a dry sample stream? How does the in-flight calibration dynamic range compare to the observations? How is LOD determined? What about breakthrough at higher VOC concentrations? What range of VOCs to the author's realistically expect to be able to measure such highly volatile compounds (e.g., ethane, ethene, ethyne), high molecular weight species (e.g., C9 aromatics), or highly polar species (e.g., butanal)? What are the current sampling limitations of this instrument?

Dry (0% relative humidity) and humid (100% relative humidity) calibrations are now included in manuscript. We show that the whole sampling system is linear up to 3ppb and we have added these results in the revised version of the manuscript.  We now include the information from all the questions above

    a.  The calibrations are now illustrated and the CAD drawings can be found in the supplementary material (Supplementary Fig. S1).

    b.  During flight, we used calibration points 0-0.5ppb that are closer to the ambient levels of the investigated species.

    c.  The LOD was determined as 3 times the std of the blank sample and considers 3 different concentration. This is now included in the text.

    d.  Higher concentrations than 3ppb were not expected in the upper troposphere for any of the species measured.

    e.  Unfortunately these species are not measured by SOFIA.

Technical comments:
- The abstract should include a description of the analytical range of the instrument including the types of compounds that can be measured.
Hydro- (> C5) and halocarbon data from SOFIA are compared

- MuPo is not defined until page 4 (P4), but first appears on P3. I suggest getting rid of the "MuPo" nomenclature entirely as it is unnecessary jargon – it's a valve. The vast majority of GC-MS systems have these types of valves and I have never heard it called a "MuPo."
Following the suggestion by the reviewer, the term "MuPo" is now replaced with "Valve".

- P4L15: It should be stated clearly that the ambient air is also a possible gaseous input to the sample selector valve.
Stated as "(including ambient air)"

- P4L28: Is the calibrated volume evacuated prior to sampling? It's not clear if the sample is pushed or pulled through the cold traps. Is the calibrated volume evacuated and therefore acts to pull the sample through the traps since the sample pump is isolated from the sample stream during sampling? Or, does the inlet pump create enough pressure to push the sample through the traps and pressurize the calibrated volume to a set pressure?
The calibrated volume is pressurized at each sample and then the pressure is released awaiting for the next sample. We hope that the supplementary CAD drawings clarify the processes followed.

- P11L35: The NOAA GC-MS system (Lerner et al., AMT 2017) also utilizes a Stirling cooler. Depending on the cooler's lift capacity, Stirling coolers can easily obtain minimum temperatures below that of liquid nitrogen (i.e., -200C). The cooling rates obtained for the NOAA GC-MS trapping system are in the range of – 6 to -10 C/second for a larger, dual-channel system.
We thank the reviewer for the comment. This reference was added.

- Figure 1 (F1) is hard to read and decipher, partly because the resolution is too low. Additional corrections/suggestions for Figure 1 are below:
F1a.) I am confused about the purpose of the 6-port inlet valve. No matter what position the valve is in, the Zero/Calibration unit will always be in the sample stream, never bypassed. The only thing that changes is whether the sample pump is upstream or downstream of the

Zero/Calibration unit. I'm assuming this has something to do with managing the hot catalyst exhaust, but it's not clear in the diagram or text.
We hope that the supplement and additional text clarifies the confusion.

F1b.) Is there a four-way union directly downstream of the inlet valve and upstream of the catalyst? If so, wouldn't the catalyst represent a large dead volume as it is always exposed to the sample stream?
You are correct but the dead volume is only about 20cc and it is fully flushed very fast. The three way valve is used to switch between the catalyst (zero measurements or calibrations) and ambient samples.

F1c.) Why is there an arrow to the right of the N2? Shouldn't the flow be going the other way to flush the catalyst when not exposed to ambient air? If so, this is not described in Section 2.1.1.
 You are correct. This is a mistake on the drawing and the arrow is now placed in the appropriate position.

F1d.) The authors missed a few of the sample line crossover junctions (e.g., just below the sample selector valve and just below HALO exterior).
We thank the reviewer for noticing this detail. Figure 1 was revised accordingly.

F1e.) I would suggest overlaying the arrows indicating flow direction over the various connections on the valve itself as well as clearly stating which position the valve is in (i.e., inject or transfer).
F1f.) There should also be a key to clearly identify 2-way, 3-way, needle valves, etc.

We believe that the supplementary drawings in increased resolution are illustrating all the sampling stages and clarify all the aforementioned questions.

- Figure 2. It is difficult to know what each part is without added annotations.
Annotations are now included in Fig. 2 (Fig. 3 in the revised version).

**Reviewer #4**

The manuscript by Bourtsoukidis et al. describes a new in-situ fast GC/MS system. The instrument adds interesting new engineering developments and is a useful addition to the suite of instruments now available for airborne trace organic gas analysis. I agree with the other reviewers comments and don't want to repeat too much here. The most essential improvements would be 1) focus more detail on the optimization and testing done to determine performance; 2) reduce the discussion of the Asian monsoon outflow (a time series of trace gases would be sufficient to demonstrate the performance in actual flight research mode); and 3) improve the clarity of the figures (the CAD drawings don't really help understand the functional design of components; the flow diagram needs to be clearer).
We thank the reviewer for acknowledging the engineering developments of our system and for providing us with feedback that will improve the quality of the manuscript.
1) More optimization tests (dry-wet calibrations, humidity alterations, ozone dependencies) have been added in the revised manuscript.
2) We have removed Figures 7 and 8 but we would suggest keeping the rest of the analysis presented. We believe that it is essential for a new instrument to demonstrate its performance in field conditions and produces high quality and comparable data.

Beyond these major revisions and those comments of the other reviewers, I had a few other questions/comments:

P3, L29: The inlet needs a complete description. The reference by Wendisch has no detail on the design, testing, and use of the inlet for trace gas analysis. This is a critical part of the system, and as far as I can tell from the flow diagram, calibration and zero additions are done downstream of the inlet. Thus, the testing that was done (or not) to determine inlet effects under ambient type conditions is important for the performance of the overall instrument.

The aircraft inlet is now included in a new figure along with the description of the flow paths and the material. Unfortunately testing of the inlet in field conditions was not possible. This possibility will be incorporated into the inlet on the next deployment in 2018.

2.1.1. Cal/Zero control. Would be interested to know if the lines with zero/cal gases are flushed continuously. Intermittent sampling of a calibration tank could cause artifacts.

The lines are not flushed continuously and indeed the system operates with an intermittent sampling of a calibration tank. Nonetheless, we did not observe artifacts such as anomalously high or low calibrations during the switch from calibration gas to ambient and back to calibration gas.

2.2.3 Liquid nitrogen container description might be better discussed prior to section 2.2.2 on Trap temperature control.

Following the suggestion by the reviewer this section is described prior to Trap temperature control.

2.2.4. This section on pressure control is not really clear at all

We hope that the new schematic of this component (illustrated in the new Fig.4) and the additional text will make it clearer for the reader. ("*Pressure regulator of liquid nitrogen. The reference pressure volume is isolated by the purple aluminium cover. Gaseous N2 that evaporates from LN2 dewar enters the bottom right inlet and once the pressure is higher by 50mbar compared with the reference pressure, the membrane (placed between the blue and purple part) is regulated by the green spring and the air stream is directed to the exaust (bottom left tube).*"

P7 L25. Were different dwell times tested, or was the effect of total dwell time of multiple ions tested? Basically, it would be important to know exactly how these parameters impacted both signal to noise (from dwell time) and precision/accuracy (from measurements across the chromatographic peak).

The different dwell times were tested and the results are now included as following: "*The dwell time of 10ms was selected as the minimum dwell time that the chromatographic peaks were well shaped. Tests have been conducted in comparing 25ms, 10ms and 5 ms. With 25ms the peaks are not clearly shaped and hence larger uncertainties are induced during peak integration. Since with 5ms dwell time the sensitivity of the detector's response was only slightly reduced (≈5%), we recommend the use of 5ms for future applications of the system.* "

Other questions: Ambient water variations: Often inlets demonstrate artifacts when the aircraft crosses from very dry to very moist conditions. I'd be interested to see examples of instrument performance (esp. for OVOC) under alternating conditions of wet and dry air. Calibration method is unclear. Could this be described in more detail, and also describe an actual sequence of sample/zero/sample/calibration/sample etc. that is used during research flights.

Following the suggestion by the reviewer all this information is now included in the manuscript. In particular, we now include wet and dry calibrations that have been performed sequentially and by altering the humidity from very dry to very moist conditions and we elaborate further on both calibrations and sequence.

P9 L 10. Measured variation of CCl4 is slightly more than 10%, and the bulk of this variation should be from the instrument, not the ambient CCl4. A precision of 10% seems high. Is this typical?

Unfortunately yes, this was typical for CCl4 during the in-flight conditions of the OMO campaign. Subsequent improvements to the system, in particular the injector and oven temperature controls, will improve this in future.

Table 1. Could you explain the value of 5% for uncertainty in sample volume? This seems too high given the quality of pressure and temperature measurements.

The uncertainty of 5% is the upper limit uncertainty which includes the uncertainty of the volume of the tank in addition to temperature and pressure measurements. However the reviewer is correct and the sample volume has an uncertainty of about 2%. The text and Table 1 were modified accordingly.

---

## Author Comment (AC2) · 13 Oct 2017

Please find our response in the attachment.

Please also note the supplement to this comment:
https://www.atmos-meas-tech-discuss.net/amt-2017-201/amt-2017-201-AC2-supplement.pdf

---

## Author Comment (AC3) · 13 Oct 2017

Please find our response in the attachment.

Please also note the supplement to this comment:
https://www.atmos-meas-tech-discuss.net/amt-2017-201/amt-2017-201-AC3-supplement.pdf

---

## Editor Decision (ED1)

Dear Authors,
A few relatively minor points to be addressed.  A few comments  below.

Abstract:
Editor comment:  Agree with reviewer - abstract should mention the analytical range (compounds measured) of the instrument.

Reviewer: P3, L30 – "In my experience, heated teflon line can be quite permeable, how do you test for contamination? e.g. TOGA has zero air introduced right at the start of the inlet as it enters the cabin."

> Response by authors: The contamination was tested in a similar manner. On board SOFIA a N2 bottle (6.0) was frequently used for diagnostic purposes. This information has been now in the text.

> Editor comment: How do the inlet blanks look? This would be nice additional information. Perhaps a sentence or two describing this is in order.

Section 2.8 9/35: Prior to each mission (or flight?)

Section 2.8 9/35: The instrument was turned on (power consumption ≈1000 W) and all gas cylinders were opened.

(Please look for other such errors in manuscript. I ran across a few of them but did not document all of them)

Editor comment: Reviewer 3 asked for information regarding the start-up time for the instrument until it is ready for prime time. I agree that this is very useful information for an aircraft instrument and should be included.

Section 2.8 10/8: (seems that this paragraph should come first in this section)
"The system was installed on board the HALO aircraft after its final configuration and certification. At first, a limited number of compounds (11) was monitored in order to ensure reliable quantification (Table 1). At the start of the OMO campaign, high sampling flows (100 sccm) resulted in inefficient water removal and hence  poor and non-reproducible chromatographic peaks. The solution was to operate the system with a lower sampling flow (40sccm) and only at high altitudes where low dew point temperatures do not affect the sampling procedure. The sampling time was 1min so a total volume of 40±6  ml was collected into the traps. During sample collection, the water trap temperature was set to -30±0.3 oC and the enrichment trap to -140±4oC. During the sample transfer, the cryofocus trap was set to -160±1oC. All traps were then heated to 120oC to ensure that all  10 volatiles were desorbed efficiently from each trap."

Editor comment: Is this still the case that the system can be operated only at high altitudes? If so, how high?

Figure 2: The inlet figure is great but not described in the text. Please describe it.

2.7.2 9/25: This increase can be attributed to production by ozone reactions occurring with other  25

species that are present in the multicomponent gas standard
 – how do you know this is true? – did you try zero air + ozone?

---

## Author Response (AR2)

Dear Authors,
A few relatively minor points to be addressed. A few comments below.

Abstract:
Editor comment: Abstract should mention the analytical range of the instrument and compounds that can be measured.

Response by authors: We have added the following sentence in the abstract:
" *Here, we present a new System for Organic Fast Identification Analysis (SOFIA), which is a custom built fast Gas Chromatography – Mass Spectrometry (GC-MS) system with a time resolution of 2-3 min and the ability to quantify atmospheric mixing ratios of halocarbons (e.g. chloromethanes), hydrocarbons (e.g isoprene), oxygenated VOCs (acetone, propanal) and aromatics (e.g. benzene, toluene, butanone) from sub-ppt to ppb levels."*

Reviewer: P3, L30 – "In my experience, heated teflon line can be quite permeable, how do you test for contamination? e.g. TOGA has zero air introduced right at the start of the inlet as it enters the cabin."

Response by authors (on AMTD): The contamination was tested in a similar manner. On board SOFIA a N2 bottle (6.0) was frequently used for diagnostic purposes. This information has been now in the text.

Editor comment: How do the inlet blanks look? This would be nice additional information. Perhaps a sentence or two describing this is in order.

Response by authors: We have added the following sentence at the end of section 2.1.1:
"*Blanks that have been obtained in such manner contain no peaks unless a leak or contamination of the inlet is present. Therefore the $N_2$ cylinder has proven to be a reliable addition for both inlet testing and standby operation*".

Section 2.8 9/35: Prior to each mission (or flight?)
Response by authors: Changed to "*flight*".

Section 2.8 9/35: The instrument was turned on (power consumption ≈1000 W) and all gas cylinders were opened.
Response by authors: Thank you for the correction.

Editor comment: Reviewer 3 asked for information regarding the start-up time for the instrument until it is ready for prime time. I agree that this is very useful information for an aircraft instrument and should be included.
(Please look for other such errors in manuscript. I ran across a few of them but did not document all of them)
Response by authors: In our response to Reviewer 3 we have chosen to report the individual times needed along the startup procedure. The start-up time can be as low as 30 min but the exact start-up time until the instrument can produce quality data can vary. We have now included the following sentence to the section 2.8:
"*The minimum start-up time required is as low as 30 min but usually a full hour was required in order to produce reliable chromatograms, clean blanks and stable calibrations.*"

Section 2.8 10/8: (seems that this paragraph should come first in this section)

"The system was installed on board the HALO aircraft after its final configuration and certification. At first, a limited number of compounds (11) was monitored in order to ensure reliable quantification (Table 1). At the start of the OMO campaign, high sampling flows (100 sccm) resulted in inefficient water removal and hence poor and non-reproducible chromatographic peaks. The solution was to operate the system with a lower sampling flow (40sccm) and only at high altitudes where low dew point temperatures do not affect the sampling procedure. The sampling time was 1min so a total volume of 40±6 ml was collected into the traps. During sample collection, the water trap temperature was set to -30±0.3 oC and the enrichment trap to -140±4oC. During the sample transfer, the cryofocus trap was set to -160±1oC. All traps were then heated to 120oC to ensure that all 10 volatiles were desorbed efficiently from each trap."

Response by authors: Thank you for the suggestion. This paragraph in now first in this section.

Editor comment: Is this still the case that the system can be operated only at high altitudes? If so, how high?

Response by authors: This solution was chosen during the campaign due to the lack of time for in-situ tests as we wanted to minimize the risk of allowing water vapor to enter the detector. Post campaign tests have shown that under such flows the water vapor is efficiently retained by the trap. We have added the following sentence at the end of the aforementioned paragraph:

"*Post campaign tests (see section 2.7) showed that under such small sampling flows the water trap sufficiently retain water vapor and therefore the instrument can be operated even under ground conditions.*"

Figure 2: The inlet figure is great but not described in the text. Please describe it.

Response by authors: We have added the following sentences in the main text:

"*Air was drawn into the aircraft through a forward-facing Trace Gas Inlet (TGI; Enviscope GmbH) (**Fig 2**, Wendisch et. al., 2016). The TGI body is constructed by aluminium (WL.-Number 3.4364, T7351) and continuously heated to 40ºC. Inside the TGI, PFA tubes (1/2' for the horizontal external tube and ¼' for the perpendicular tube) are used for the air streams. The main air flow is parallel to the flight direction and the inlet pump of SOFIA draws air from the T-piece inside the TGI, perpendicular to the flight direction. In the cabin,…*"

2.7.2 9/25: This increase can be attributed to production by ozone reactions occurring with other 25 species that are present in the multicomponent gas standard – how do you know this is true? – did you try zero air + ozone?

Response by authors: Yes we did as described in a different paragraph (P9 L12-15). In order to avoid similar misconceptions we have added the following sentence:

"*This increase can be attributed to production by ozone reactions occurring with other species that are present in the multicomponent gas standard, as the supply of ozone enriched zero air did not result in production of OVOCs.*"

Additional comment by the authors: In the acknowledgment section we have added the following sentence: "*We thank Eric Apel and Aaron Johnson for helpful discussions during the preliminary design phase.*"

Final remark by the authors: We would like to thank the editor for choosing suitable reviewers and for providing us with insightful feedback. We believe that the very fruitful discussions and revisions suggested during the review process have substantially improved the quality of our manuscript.